# One-Step Offline Distillation of Diffusion-based Models via Koopman Modeling

**Nimrod Berman**[*1]    **Ilan Naiman**[*1]    **Moshe Eliasof**[1,2]    **Hedi Zisling**[1]    **Omri Azencot**[1]

[1]Ben-Gurion University of the Negev    [2]University of Cambridge

https://sites.google.com/view/koopman-distillation-model/

## Abstract

Diffusion-based generative models have demonstrated exceptional performance, yet their iterative sampling procedures remain computationally expensive. A prominent strategy to mitigate this cost is *distillation*, with *offline distillation* offering particular advantages in terms of efficiency, modularity, and flexibility. In this work, we identify two key observations that motivate a principled distillation framework: (1) while diffusion models have been viewed through the lens of dynamical systems theory, powerful and underexplored tools can be further leveraged; and (2) diffusion models inherently impose structured, semantically coherent trajectories in latent space. Building on these observations, we introduce the *Koopman Distillation Model (KDM)*, a novel offline distillation approach grounded in Koopman theory - a classical framework for representing nonlinear dynamics linearly in a transformed space. KDM encodes noisy inputs into an embedded space where a learned linear operator propagates them forward, followed by a decoder that reconstructs clean samples. This enables single-step generation while preserving semantic fidelity. We provide theoretical justification for our approach: (1) under mild assumptions, the learned diffusion dynamics admit a finite-dimensional Koopman representation; and (2) proximity in the Koopman latent space correlates with semantic similarity in the generated outputs, allowing for effective trajectory alignment. KDM achieves highly competitive performance across standard *offline distillation* benchmarks.

## 1 Introduction

Diffusion-based generative models [72], including score-based and flow-matching variants [30, 47], have become ubiquitous across a wide range of domains—from images and videos to audio and time series [18, 31, 41, 16, 59, 21, 26]. These models now surpass approaches such as GANs and VAEs in terms of sample quality, while also exhibiting greater training stability. Despite these advantages, one of their key limitations remains the high computational cost associated with sampling [63, 68, 51]. Generating a single high-fidelity sample typically requires executing a lengthy iterative process, progressively refining random noise through dozens or even hundreds of model evaluations [30].

A widely adopted strategy to address this involves minimizing the number of inference steps [73], for example by leveraging improved numerical solvers or designing more expressive noise schedules, enabling high-quality generation with significantly fewer denoising iterations. Another increasingly popular alternative is *distillation* [68], where a student model learns to emulate the behavior of a teacher model. Distillation approaches vary in supervision and setup: in online distillation, the student directly learns from the teacher's predictions during training, with the teacher providing trajectory information across time steps. In contrast, *offline distillation* relies on precomputed noise–image pairs or teacher-generated samples only, where the student learns without any further access to the teacher model, including its weights, evaluations, or internal states.

---

[*]Equal contribution

39th Conference on Neural Information Processing Systems (NeurIPS 2025).

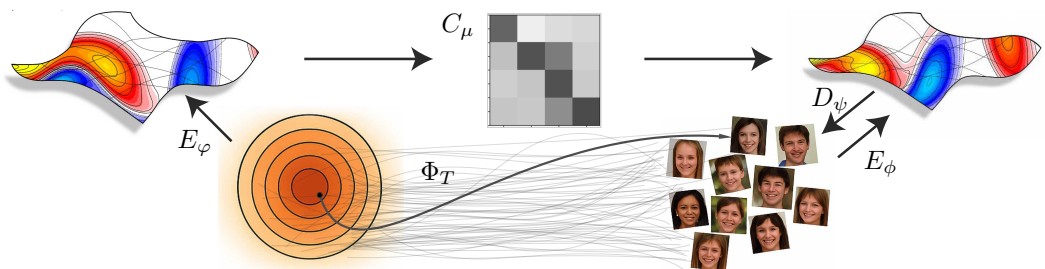

Figure 1: A Gaussian sample (bottom left) evolves into a clean image (bottom right) via nonlinear dynamics $\Phi_T$. Leveraging the Koopman framework, learned encoders $E_\varphi$ and $E_\phi$ transform noise and image into an embedding space where the evolution becomes linear under $C_\mu$ (top).

Offline methods inherently offer key advantages over online tools. Training on precomputed, teacher-generated noise–image pairs avoids multi-stage training and repeated teacher queries, reducing both neural function evaluations and memory footprint [24]. It also enables pre-filtering or anonymization of potentially privacy violations [22]. Consequently, once the dataset is produced, the teacher can be discarded, shrinking the attack surface and supporting secure collaboration (e.g., outsourcing training without exposing proprietary information) [12, 32, 61]. Beyond privacy and efficiency, offline distillation is model-agnostic and architecture-general (as evidenced in our experiments), and it scales flexibly with parameter budgets. On the limitation side, offline setups require upfront generation and storage costs for the precomputed corpus and, lacking teacher access beyond the noise–image pairs, may constrain model expressivity and adaptability. By contrast, online methods may warm-start from teacher weights or query the teacher for stepwise supervision. Despite these trade-offs, offline distillation remains a practical, private, and sustainable approach to generative-model distillation. The two paradigms can be considered complementary, and the optimal choice depends on factors such as privacy requirements, computational cost, and deployment constraints.

Towards developing a distillation method, we identify two key observations that guide our approach: (1) While denoising-based methods leverage dynamical systems perspectives for both general generative modeling [76, 47] and distillation [89, 13], a wide range of powerful tools from dynamical systems theory remain underexplored; (2) Trained diffusion models impose a structured organization on the latent space, coherently mapping noisy inputs toward clean data. Specifically, our empirical analysis suggests that semantically related images tend to originate from nearby points in the noise space.

Building on these observations, we propose leveraging a dynamical systems approach that has not yet been explored in the context of distillation: Koopman operator theory. This theory is a classical yet recently revitalized approach for representing nonlinear systems with linear dynamics in an embedded space [42, 67]. The core idea is that by mapping the system into a suitable embedding space, its evolution becomes linear. Thus, learning the diffusion process then reduces to jointly learning these embeddings and the linear operator. While Koopman operators are typically infinite-dimensional, we prove that an exact finite-dimensional representation exists under mild assumptions [33]. Moreover, prior work suggests that the Koopman operator preserves the structure inherent to certain dynamical systems [28, 9]. This motivates the use of Koopman theory to harness the observed coherent semantic structure in diffusion dynamics, offering a principled foundation for accurate teacher model distillation. To this end, we formally connect the observed structure in diffusion models to the Koopman framework by proving that this structure is preserved within the learned dynamical representation.

In practice, we propose KDM (*Koopman Distillation Model*), a framework for modeling diffusion processes through linear evolution in a learned embedding space. Given a noisy input, an encoder maps it into an embedding space where a learned Koopman operator evolves it forward, and a decoder reconstructs the clean sample. A schematic illustration of our framework is presented in Fig. 1. The training objective encourages semantic preservation (via reconstruction loss), accurate latent linear dynamics (via Koopman consistency), and fidelity in output space (via prediction and adversarial losses). This design naturally yields fast, single-step generation, while preserving structural coherence distillation. We evaluate our method on the offline distillation benchmark, while adding two higher-resolution datasets, and achieve highly competitive results in both unconditional

and conditional generation. These results demonstrate the effectiveness of our Koopman-based formulation in bridging the performance gap with online methods, offering a fast and high-fidelity offline diffusion model distillation. **Our key contributions are:**

1. *Observing diffusion dynamics.* We identify, analyze and formalize two key observations: (i) tools from dynamical systems theory, specifically Koopman operator theory, remain underexplored in diffusion modeling and distillation; (ii) training induces coherent semantic organization in noise space. These insights motivate a structure-promoting dynamics-aware approach to distillation.

2. *A principled and practical Koopman-based distillation framework.* We develop a theoretical foundation proving the existence of finite-dimensional Koopman representations and semantic structure preservation under mild assumptions and introduce a simple, scalable encoder–linear-dynamics–decoder architecture that preserves the teacher generation dynamics semantic structure.

3. *Highly competitive results.* We conduct a comprehensive unconditional and conditional evaluation showing that our method outperforms prior offline distillation approaches, achieving FID improvement, while enabling efficient, single-step generation.

## 2 Related Work

**Generative Modeling.** Deep generative models have made significant progress in recent years, with diffusion models emerging as a leading framework for high-quality image, audio, and molecular generation. Building on principles from nonequilibrium thermodynamics and score matching [72, 76], these models define a forward process that gradually corrupts data with noise and learn to reverse it via a neural network-based denoising process. Variants like DDPMs [30] and score-based generative models [76] have shown competitive performance across a wide range of benchmarks, often surpassing VAEs and GANs [40, 27] in sample fidelity and diversity.

**Accelerating Diffusion Sampling.** Diffusion models generate high-quality samples but are slow due to iterative sampling. One approach improves integration using higher-order solvers and expressive noise schedules [73, 49], achieving high-quality results in significantly fewer steps. Another class of approaches seeks to distill a student model from a pre-trained teacher, reducing the sampling process to a single or few steps. Online distillation techniques [50, 68] supervise the student directly with teacher predictions during training, but require continuous access to the full teacher model, resulting in high memory and compute overhead. Recent methods demonstrate high-quality one-step generation across CIFAR-10, ImageNet, and text-to-image benchmarks [86, 93], using objectives based on distribution matching [86], score identity [93], adversarial training [69, 71], and consistency models [75, 38] among others [13, 52, 62, 94, 6]. Offline distillation methods [24] rely on teacher-generated noise-data pairs and decouple student training from the teacher. Aside from the concurrent work [82], current distillation methods largely unexplored Koopman-based perspectives and their relation to the teacher's underlying dynamics.

**Koopman-Based Modeling.** Koopman operator theory provides a powerful framework for modeling nonlinear dynamical systems via linear operators acting on observable functions [42, 67]. This view has been applied to tasks such as deep learning of dynamical systems [79, 53, 84, 65, 58, 60], sequential disentanglement [7, 5], and control [29]. A related strand of work focuses on identifying coherent sets—regions of the state space that evolve coherently over time—using Koopman-based embeddings [56, 23]. While these works exploit the Koopman structure for interpretability or stability, our work focuses on using Koopman-based modeling for distilling diffusion models. In this context, we demonstrate that the learned embedding space in diffusion models encodes semantically meaningful structure, and we leverage this to design a principled offline distillation framework.

## 3 Mathematical Background

**Deterministic Diffusion Sampling.** We consider deterministic sampling in diffusion models, as introduced in the EDM framework [35]. A pre-trained diffusion model defines a time-indexed denoising vector field $f_\theta : \mathbb{R}^n \times [0, T] \to \mathbb{R}^n$, which maps a noisy latent state $x_t \in \mathbb{R}^n$ at time $t \in [0, T]$ toward the data manifold (at time 0). Sampling is performed by solving the reverse-time ordinary differential equation (ODE),

$$\frac{\mathrm{d}\, x_t}{\mathrm{d}\, t} = f_\theta(x_t, t)\,, \quad x_T \sim \mathcal{N}(0, \sigma_T^2 \mathbf{I})\,, \tag{1}$$

integrated backward from a Gaussian sample $x_T$ at time $T$ to a clean sample $x_0$ at time $t = 0$. The function $f_\theta$ is typically trained to approximate the score function $\nabla_{x_t} \log p_t(x_t)$ or a reparameterization thereof, such as the probability flow [76]. These deterministic formulations enable sampling via high-order ODE solvers and provide a tractable setting for distillation and trajectory analysis [35].

**Koopman Operator Theory.** Let $\Phi_t : \mathbb{R}^n \to \mathbb{R}^n$ define a (possibly time-dependent) dynamical system mapping states forward in time, i.e., $x_t = \Phi_t(x_0)$. The Koopman operator $\mathcal{K}_t$ [42] is a linear operator that acts on scalar-valued *observables* $g : \mathbb{R}^n \to \mathbb{C}$ via composition,

$$\mathcal{K}_t g := g \circ \Phi_t \, , \tag{2}$$

mapping each observable to its pullback along the dynamics. While the original system $\Phi_t$ may be nonlinear, $\mathcal{K}_t$ is linear on the space of observables $L^2(\mathbb{C}^n)$ (with respect to a suitable measure) [20]. In classical Koopman theory, this operator is infinite-dimensional. However, if there exists a finite-dimensional subspace $\mathcal{O} = \text{span}\{g_1, \ldots, g_d\} \subset L^2(\mathbb{C}^n)$ that is invariant under $\mathcal{K}_t$ for all $t \in [0, T]$, then the dynamics of $\Phi_t$ can be exactly represented in this lifted space via a family of linear operators $C_t \in \mathbb{R}^{d \times d}$, $t \in [0, T]$, such that for all $g \in \mathcal{O}$, $\mathcal{K}_t g = C_t g$. This allows modeling nonlinear dynamics through linear evolution in a learned observable space, a principle that underpins our Koopman-based distillation framework.

## 4 Koopman Distillation of Diffusion Models

We propose *KDM*, a principled offline distillation framework for diffusion models based on Koopman operator theory. The key idea is to model the dynamics of the generative process, specifically the transformation from a noisy sample $x_T$ to its clean counterpart $x_0$, as a single-step *linear* evolution in a learned embedding space. This approach enables fast and structure-preserving generation in a single forward pass, without requiring access to the full diffusion trajectory or teacher model at training time.

**Dynamical System Perspective on Diffusion Models.** We build upon two key observations regarding diffusion models. *First*, diffusion can be understood as a dynamical system, where the evolution of a noisy data point $x_t$ is governed by a time-dependent velocity field, see Eq. (1). This continuous-time formulation admits an integrated flow map $\Phi_t$ such that $x_t = \Phi_t(x_0)$, where $\Phi_t$ evolves the initial condition $x_0$ forward in time. This formulation makes it natural to propose adopting a Koopman operator perspective, a novel approach in the context of distillation, which enables the lifting of nonlinear dynamics into a linear evolution in function space under suitable choices of observables, see Eq. (2).

The *second* observation considers the structure of the trained diffusion dynamics. Score-based diffusion models learn the score $\nabla_{x_t} \log p_t(x_t)$ of noise-corrupted data at multiple noise levels and then reverse that specific corruption process using Langevin dynamics [76] or learned reverse kernels [30] to transform Gaussian noise into images. This framework has been unified through the lens of stochastic differential equations (SDEs) by [76]. Flow matching models [47] learn a continuous velocity field that transports a simple reference distribution directly into the data distribution, where sampling is typically performed by integrating the learned ordinary differential equation (ODE). In what follows, we study the structure of the dynamical mapping arising from diffusion-based learning.

We pose the following question: *Is there semantic structure in the generative process of diffusion models?* In general, prior to training, there is no assurance of semantic structure between the noise and data distributions (see Fig. 2, left). However, after training, we observe the emergence of a coherent and semantically meaningful mapping from the Gaussian noise distribution to the data distribution (see Fig. 2, right). Specifically, by utilizing a simple 2D checkerboard toy dataset, we train an EDM [35] model and generate 50,000 samples, recording both the initial noise vectors and their corresponding generated outputs on the checkerboard. We then color-code each sample by its target checker cell and assign the same color to its originating noise vector. This visualization reveals that samples originating from the same semantic region in data space tend to cluster together in noise space, indicating structured and coherent dynamics. Notably, we observe a similar pattern in both flow matching model and our KDM. This emergence of coherent structure, together with prior evidence that the Koopman operator preserves structure in certain dynamical systems [28, 9], highlights the relevance of Koopman theory to diffusion dynamics. Indeed, our theoretical analysis (Sec. 5) establishes a formal link between this observed structure and Koopman theory, motivating its adoption as a principled framework for modeling these dynamics.

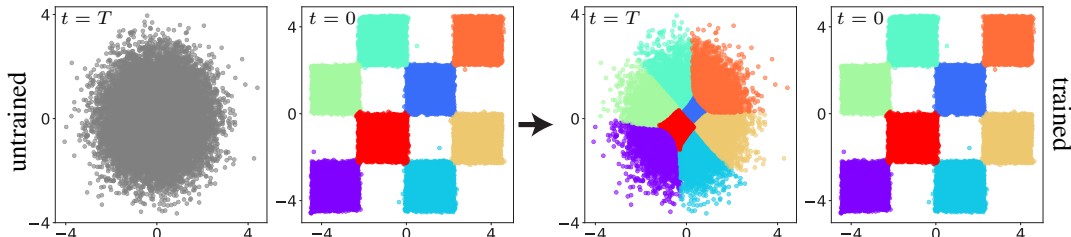

Figure 2: Untrained noise distributions are semantically unstructured (left). Training EDM on the data reveals the emergence of coherent clusters in noise space, i.e., at time $T$ (right).

**Koopman-based Distillation.** Our Koopman distillation framework is designed by the following principle: given pairs of noisy and clean data $(x_T, x_0)$, where $x_0 = \Phi_T^{-1}(x_T)$ for some unknown nonlinear map $\Phi_t^{-1}$, we seek to model this transformation using a finite-dimensional linear operator in an embedding space, following Koopman theory. To achieve this, we introduce observable functions $\{g_1, g_2, ..., g_d\}$, parameterized as *encoder networks* $E_\phi, E_\varphi$ for $x_0, x_T$, respectively, that map states from the original data space into a latent observable space. In this space, we assume the dynamics can be modeled linearly by a finite-dimensional Koopman operator linear layer, $C_\eta$. Finally, a corresponding *decoder network* $D_\psi$ maps the evolved observables back to the data space. The overall modeling objective becomes:

$$x_0 \approx D_\psi(C_\eta E_\varphi(x_T)) . \tag{3}$$

This formulation distills the potentially complex, nonlinear generative process into a simple one-step linear map in the embedded observables space, while maintaining fidelity to the original dynamics.

**Koopman's Loss Functions.** In what follows, $d(\cdot, \cdot)$ denotes a suitable distance function, such as mean squared error (MSE) or a perceptual loss (e.g., LPIPS [88]). To effectively learn the components $E_\phi, D_\psi$, and $C_\eta$, we define a composite loss comprising three terms (see similar objectives in [53, 4]):

$$\mathcal{L}_{\text{Koopman}} = \mathcal{L}_{\text{rec}} + \mathcal{L}_{\text{lat}} + \mathcal{L}_{\text{pred}} . \tag{4}$$

The first term, the *reconstruction loss* $\mathcal{L}_{\text{rec}}$, ensures that the encoder-decoder pair $(E_\phi, D_\psi)$ can faithfully reconstruct $x_0$:

$$\mathcal{L}_{\text{rec}} = d\big(x_0, D_\psi(E_\phi(x_0))\big) . \tag{5}$$

This term regularizes the embedding by encouraging $E_\phi$ to preserve sufficient semantic information about the data, while not encoding any dynamic features. The second term, the *latent dynamics loss* $\mathcal{L}_{\text{lat}}$, enforces consistency of the linear evolution in the observable space:

$$\mathcal{L}_{\text{lat}} = d\big(E_\phi(x_0), C_\eta E_\varphi(x_T)\big) , \tag{6}$$

where $C_\eta$ is the Koopman operator in latent space. This term ensures that applying $C_\eta$ to the embedded noisy state $E_\varphi(x_T)$ approximates the embedded clean state $E_\phi(x_0)$. $C_\eta$ is either implemented with a regular learning linear layer or by a new approach we develop that factorizes the matrix, which enables eigenspectrum control (KDM-F). We describe the decomposition and its merits in App. C.

Finally, the *prediction loss* $\mathcal{L}_{\text{pred}}$ evaluates the full end-to-end prediction after encoding, applying the Koopman operator, and decoding:

$$\mathcal{L}_{\text{pred}} = d\big(x_0, D_\psi(C_\eta E_\varphi(x_T))\big) , \tag{7}$$

encouraging the system to accurately predict $x_0$ from $x_T$ in a single forward pass through the Koopman pipeline. Together, these terms provide complementary supervision signals.

**Adversarial Loss.** Adversarial frameworks are widely employed in distillation methods [70, 38, 77, 78]. For example, [38] leverages an adversarial loss to enhance the fidelity of one-step generation. Motivated by these approaches, we introduce an auxiliary adversarial loss to further refine the distillation process. Specifically, we introduce a discriminator network $D_\gamma$, parameterized by $\gamma$, and define the adversarial loss as:

$$\mathcal{L}_{\text{adv}}(\phi, \psi, \gamma) = \mathbb{E}_{x_0 \sim p_{\text{data}}}[\log D_\gamma(x_0)] + \mathbb{E}_{x_T \sim p_{\text{data}}}[\log(1 - D_\gamma(\hat{x}_0))] , \tag{8}$$

where $\hat{x}_0 = D_\psi(C_\eta E_\varphi(x_T))$ is the Koopman-predicted sample. Thus, the final training objective is:

$$\mathcal{L} = \mathcal{L}_{\text{Koopman}} + \lambda_{\text{adv}}\mathcal{L}_{\text{adv}} , \tag{9}$$

where $\lambda_{\text{adv}}$ balances the adversarial component. In practice, throughout our experiments, we choose $\lambda_{\text{adv}} = 0.01$. Importantly, we utilized a simple $D_\gamma$ with four convolutional layers. By enhancing the perceptual quality of the predictions without altering the fundamental Koopman modeling, this adversarial augmentation strengthens the overall generation performance while preserving the structured distillation at the heart of our method [8]. We present losses and more ablation studies in App. D.1. Additionally, A pseudo-code of our KDM is provided in App. E.1.

**Control Modeling for Conditional Dynamics.** A crucial aspect of generative modeling with denoising models is the ability to condition generation on auxiliary signals or labels [18]. Koopman theory provides a natural framework to incorporate such conditioning through control signals that influence the Koopman operator [3, 11]. Formally, let $c$ denote the control signal or conditioning variable (e.g., a class label) that we want the generative process to be conditioned on. Our goal is to model the conditional dynamics $\Phi(x_T, c)$. In the Koopman framework, we extend the observable and evolution operators to incorporate $c$, and formulate the generation as:

$$D_\psi\big(C_\eta E_\phi(x_T, c) + C_\mu(c),\ c\big) \approx \Phi(x_T, c),$$

where $C_\mu(c)$ is a control-dependent modulation term (e.g., a linear transformation of $c$). This formulation enables our model to adapt its generative trajectory based on the conditioning signal, seamlessly integrating the conditioning mechanism into the Koopman dynamics. We apply this control modeling in our conditional generation experiments.

# 5 Theoretical Properties of KDM

We now state the **first** key theoretical result, establishing the existence of an exact finite-dimensional Koopman representation under mild conditions. Specifically, for an analytic dynamical system, it is possible to approximate its nonlinear evolution to arbitrary accuracy by a finite-dimensional linear operator acting on lifted coordinates. In particular, a small number of polynomial observables can capture the system's dynamics with controllable approximation error. This result is significant because it means we can faithfully replicate the complex behavior of the teacher model using a simpler, linear system in a transformed space. In practical terms, it ensures our student can learn to follow the teacher's dynamics with high accuracy.

**Theorem 5.1** (Finite Koopman Operator for Analytic Dynamics). *Let $\Phi : \mathbb{R}^n \to \mathbb{R}^n$ be an analytic map, and let $x_T \sim \mathcal{N}(0, I_n)$. Then, for any $\epsilon > 0$, there exists a linear operator $C \in \mathbb{R}^{d \times d}$ such that*

$$\mathbb{E}\big[\|\xi(\Phi(x_T)) - C\xi(x_T)\|^2\big] \leq \epsilon , \tag{10}$$

*where $\xi : \mathbb{R}^n \to \mathbb{R}^d$ denotes the observable lifting map. Moreover, the required dimension $d$ grows at most polynomially with $1/\epsilon$, and the mapping $\Phi$ can be approximated arbitrarily well using multivariate polynomials up to degree $d$.*

The proof builds on the analytic formulation of $\Phi$, allowing a convergent multivariate Taylor expansion [81], and on exact finite-dimensional Koopman operators for polynomial dynamics [33], see App. A.

Our **second** theoretical result concerns the structure induced by the reverse diffusion dynamics, under a set of assumptions established by Thm. 5.1. Under these conditions, the theorem states that proximity in the Koopman coordinate representation at the initial noise time implies semantic similarity between the corresponding samples at data time. In practice, the theorem says that if two inputs start out close together in the transformed space, the images they generate will be semantically similar. This provides a theoretical foundation for the structure-preserving behavior of our model, as observed empirically in Sec. 6.1.

**Theorem 5.2** (Semantic Proximity via Koopman-Invariant Coordinates). *Let $\Phi_t : \mathbb{R}^n \to \mathbb{R}^n$ be the reverse diffusion flow of a trained model (from time $T$ to $0$), and let $\mathcal{O} \subset L^2(\mathbb{R}^n)$ be a finite-dimensional subspace such that:*

  *1. $\mathcal{K}_t\mathcal{O} \subset \mathcal{O}$ for all $t \in [0, T]$,*

  *2. $\exists\, C_t : \mathbb{R}^d \to \mathbb{R}^d$ such that $\mathcal{K}_t\xi = \xi \circ \Phi_t = C_t\xi$ for all $\xi \in \mathcal{O}$,*

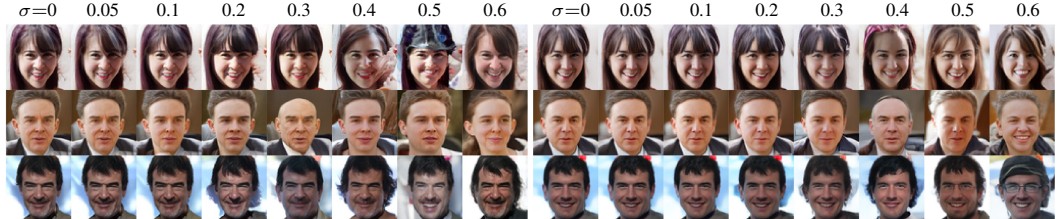

Figure 3: Visualization of vicinity comparison across different noise levels $\sigma$. Left: EDM [35]. Right: our method. Each image corresponds to a small perturbation around the original noise sample.

3. $\Phi_T$ is locally Lipschitz.

For any $x_T^1, x_T^2 \in \mathbb{R}^n$, define $x_0^j := \Phi_T(x_T^j)$ for $j = 1, 2$, and let $\xi : \mathbb{R}^n \to \mathbb{R}^d$ be the vector-valued observable $\xi(x) := (\xi_1(x), \ldots, \xi_d(x)) \in \mathcal{O}$. Then:

$$\|x_0^1 - x_0^2\| \leq L \cdot \|\Phi_T(x_T^1) - \Phi_T(x_T^2)\| \leq L \cdot \|C_T\| \cdot \|\xi(x_T^1) - \xi(x_T^2)\|.$$

The proof builds on the observation that diffusion model dynamics tend to preserve semantic similarity between inputs and outputs. By leveraging the Koopman operator's spectral properties, we construct a finite-dimensional space of functions that remains stable under the model's evolution. Within this space, the Koopman coordinates $\xi$ provide a locally bi-Lipschitz representation, while the reverse diffusion map $\Phi_T$ is Lipschitz continuous. Together, these properties ensure that semantically similar inputs remain close throughout the diffusion process. Note that the theorem holds also for $\xi := \text{id}$, i.e., the identity map [54, 19]. The full proof is given in App. B.

Importantly, we observe in practice an average spectral norm of the Koopman operator $C_T$ of $0.88 \pm 0.18$, indicating a reasonably tight bound and supporting the practical utility of our theory. Moreover, our framework enables explicit control over this norm via eigenvalue penalties on the Koopman operator. As detailed in App. C, we factorize $C_T$ to permit direct constraints on its spectrum (e.g., radius or banding), providing a principled regularizer on $\|C_T\|$.

## 6 Experiments

We begin by further investigating the emergent dynamics structure that map noise to images, and its presence in KDM (Sec. 6.1). We then extensively assess our method on the offline distillation benchmark (Sec. 6.2), including analyses of computational complexity, data and parameter scaling, and denoiser-agnostic utility. Additionally, an ablation study of the loss functions and adversarial losses stability is provided in App. D.1. Our code is in `https://github.com/azencot-group/KDM`.

**Setup and evaluation.** We follow the protocol of GET [24] for generating noisy-clean pairs and for evaluation, using the CIFAR-10 [43], FFHQ 64×64 [37], and AFHQv2 64×64 [15] datasets. Our training setup matches that of PD [68] and GET [24]. For evaluation, we report image quality using Fréchet Inception Distance (FID) and Inception Score (IS), computed over 50k samples. Both encoders $E_\varphi$ and $E_\phi$, as well as the decoder $D_\psi$, are implemented as compact UNets with reduced output channels to ensure a similar parameter size to the other single-step methods. The discriminator consists of four simple Conv2D layers. Full implementation details are provided in App. E.3.

### 6.1 Observations of Emergent Latent Structure

**Real-world data dynamics: local structure analysis.** Extending the 2D toy experiment from Sec. 4, we now explore the high-dimensional FFHQ 64×64 dataset. Since dimensionality reduction can obscure fine-grained patterns, we instead analyze the local neighborhood of a single latent sample $x_T \sim \mathcal{N}(0, I_n)$. We generate perturbed versions $\hat{x}_T = x_T + \varepsilon \cdot \sigma$ with $\varepsilon \sim \mathcal{N}(0, I_n)$, and normalize each $\hat{x}_T$ using its own mean and standard deviation. This setup tests whether semantic structure emerges locally around $x_T$. We vary $\sigma \in \{0, 0.05, \ldots, 0.60\}$ and visualize the outputs in Fig. 3 (left). For all $\sigma < 0.3$, the images remain semantically aligned with the original image (when $\sigma = 0$), indicating local smoothness and manifold adherence. As $\sigma$ grows, semantics diverge but retain global coherence even at $\sigma = 0.60$. The same experiment on KDM (Fig. 3, right) reveals similar behavior,

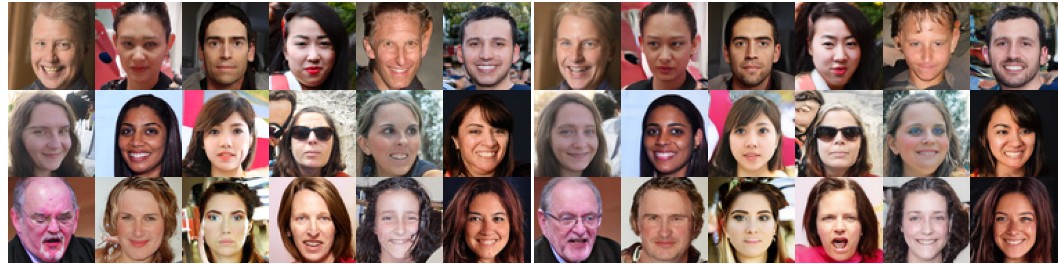

Figure 4: Sampled images using 79 NFEs with EDM [35] (left), and using 1 NFE with KDM (right).

affirming that KDM preserves the local semantic structure of the original EDM. Our results show that both models organize latent space into meaningful neighborhoods, even in high-dimensional settings.

**Visual analysis of dynamics distillation.** Beyond preserving local structure, KDM also approximately reproduces the noise-to-image mapping of EDM. When starting from the same noise vector $x_T \sim \mathcal{N}(0, I_n)$, sampled using seeds not seen during training, both EDM (with 79 NFEs) and KDM (with a single NFE) generate perceptually similar outputs (see Fig. 4). This suggests that KDM generalizes well beyond its training data and effectively distills the mapping from noise to data. We observe this consistency across multiple datasets and include further qualitative results in App. D.3. To quantify this, we generate 50k samples using both methods and compare outputs using LPIPS and MSE. KDM achieves scores of 0.13 (LPIPS) and 0.009 (MSE), in contrast to 0.48 and 0.57 when comparing randomly permuted images, demonstrating the similarity between the models' generation. To complement quantitative metrics, we conduct a human study where participants choose the more realistic image collage between our model and EDM (Fig. 4; details in App. E.4). Evaluators showed no clear preference: $55\%$ of 300 choices favored our model, highlighting its strong fidelity.

**Outliers and their relation to the dynamics structure.** We analyze a recurring failure mode in both the teacher and distilled models: the generation of outlier samples that fall outside the designated checkerboard regions. These out-of-distribution points often appear in the final outputs and reflect a limitation in capturing the intended data distribution. To investigate their origin, we trace these outliers back through the generation process and identify two primary sources: (1) the tails of the Gaussian prior and (2) regions near decision boundaries between adjacent checkerboard cells. Fig. 5C shows this behavior for the distilled model (black crosses denote outliers); additional comparisons with the teacher are provided in App. E.7. These findings suggest that the model's dynamics are more error-prone in high-uncertainty regions, offering potential directions for improvement. Integrating more robust sampling techniques or leveraging self-supervised signals may help mitigate such failures. This insight could also inform future efforts in uncertainty estimation.

### 6.2 One-step Generation Results

**Unconditional and conditional generation with CIFAR-10.** We evaluate on the standard *offline distillation* benchmark of [24], which distills a pre-trained EDM on CIFAR-10. Whereas the EDM teacher samples with 35 function evaluations (NFEs), our student attains comparable quality with a *single* NFE. We also adapt two recent SOTA baselines, IMM [91] and RF-2++ [45], to the same strictly offline setting, where only teacher-generated data are available and the original model weights (and online queries) are *not* accessible (the RF-2++ variant corresponds to its "w/o pre-train" configuration). All methods are evaluated under identical protocols. Tab. 1 reports unconditional generation results,

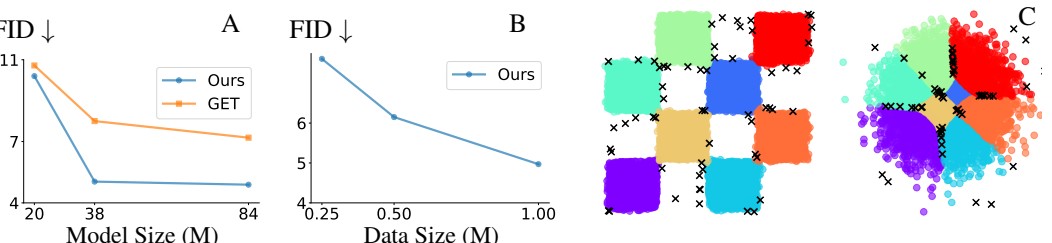

Figure 5: A) Model size vs FID. B) Data Size vs FID. C) Outlier analysis

Table 1: Generative performance on unconditional CIFAR-10.

| Models (↓) / Metrics (→) | NFE ↓ | FID ↓ | IS ↑ |
|---|---|---|---|
| *Diffusion Models* | | | |
| DDPM [30] | 1000 | 3.17 | 9.46 |
| Score SDE [73] | 2000 | 2.2 | 9.89 |
| EDM [37] | 35 | 2.04 | 9.84 |
| *Continuous Flows* | | | |
| Flow Matching (Diffusion) [46] | 183 | 8.06 | - |
| Flow Matching (OT) [80] | 50 | 4.94 | - |
| 2-rf++ [45] | 1 | 3.07 | - |
| *GANs* | | | |
| StyleGAN-XL [36] | 1 | 1.85 | - |
| *Diffusion Distillation* | | | |
| PD [68] | 1 | 9.12 | - |
| DFNO [90] | 1 | 4.12 | - |
| iCD-EDM (LPIPS) [74] | 1 | 2.83 | 9.54 |
| CTM (LPIPS) [74] | 1 | 1.98 | - |
| ECM [25] | 1 | 4.54 | - |
| **Offline Distillation** | | | |
| GET [24] | 1 | 6.91 | 9.16 |
| 2-rf++ w/o pre-train [45] | 1 | 6.32 | 9.01 |
| IMM [91] | 1 | 4.81 | 9.16 |
| KDM (EDM) | 1 | **4.65** | **9.21** |
| KDM (FM) | 1 | 5.91 | 8.86 |
| KDM-F (EDM) | 1 | **4.68** | 9.08 |

Table 2: Generative performance on class-conditional CIFAR-10.

| Models (↓) / Metrics (→) | NFE ↓ | FID ↓ | IS ↑ |
|---|---|---|---|
| *GANs* | | | |
| BigGAN [10] | 1 | 14.73 | 9.22 |
| StyleGAN2-ADA [36] | 1 | 2.42 | 10.14 |
| *Diffusion Distillation* | | | |
| Guided Dist. ($w = 0.3$) [55] | 1 | 7.34 | 8.90 |
| ECM [25] | 1 | 3.81 | - |
| **Offline Distillation** | | | |
| GET [24] | 1 | 6.25 | 9.40 |
| KDM (EDM) | 1 | **3.56** | **9.54** |
| KDM-F (EDM) | 1 | **3.24** | **9.68** |

Table 3: Generative performance on unconditional generation on FFHQ 64×64 and AFHQv2 64×64. *Indicates our reproduction.

| Models (↓) / Metrics (→) | NFE ↓ | FID ↓ | IS ↑ |
|---|---|---|---|
| *FFHQ 64×64* | | | |
| EDM* | 79 | 2.47 | 3.37 |
| KDM (EDM) | 1 | 6.54 | 3.12 |
| *AFHQv2 64×64* | | | |
| EDM* | 79 | 2.02 | 9.04 |
| KDM (EDM) | 1 | 4.85 | 8.25 |

where our method achieves highly competitive performance, surpassing the previous offline baseline in both FID and IS. In particular, we observe ≈ 25% reduction in FID over GET. Tab. 2 shows results for class-conditional generation, where our model yields ≈ 40% improvement in FID and superior IS compared to GET. Although offline distillation offers advantages in efficiency, privacy, and flexible deployment [22, 24], a performance gap with online methods remains. This may partly be because online approaches leverage the full generation trajectory, whereas, in the offline pairwise setup, methods only observe two endpoints or just the generated data, placing them at a natural disadvantage. Our results demonstrate a substantial step toward closing this gap.

Furthermore, unlike prior methods that rely on architectural assumptions [25, 45], our approach is denoiser-agnostic and successfully distills both EDM and flow matching (FM) models (see Tab. 1), demonstrating broad applicability. Finally, we implemented a factorized KDM (KDM-F). Since every matrix $C_\eta$ is diagonalizable up to an arbitrarily small perturbation of the entries [2], we can write: $C_\eta = P\Lambda P^{-1}$ where $P \in \mathbb{C}^{d \times d}$ is an invertible matrix and $\Lambda = \text{diag}(\lambda_1, \ldots, \lambda_d) \in \mathbb{C}^{d \times d}$. While enabling efficient implementation of spectral penalties on $\Lambda$ (see complexity analysis below), this approach preserves strong performance and accuracy.

**Unconditional generation with FFHQ and AFHQv2.** To assess the scalability of KDM to higher-resolution data, we evaluate our method on FFHQ and AFHQv2, distilling each into a single-step generator and assessing performance both quantitatively (Tab. 3) and qualitatively (App. D.3). Our method yields FID scores approximately 2.5× higher than EDM, similar to the degradation observed on CIFAR-10. Due to the high computational cost, we omit GET results: training a single model takes 5–7 weeks on an A6000 GPU, which is technically infeasible in our computational resources. We also exclude ImageNet, as large-scale training with EDM [35] is computationally infeasible for us.

**Scalability in parameters and data.** Offline distillation inherently supports architectural independence from the teacher model, allowing flexible scaling of the student to meet diverse deployment needs [24, 25]. To assess this, we evaluate our method against GET across three model sizes—20M, 38M, and 84M parameters (Fig. 5A). Our method consistently outperforms GET in FID scores at all

scales, with especially notable gains at higher capacities (e.g., 4.89 vs. 7.19 at 84M), demonstrating its effectiveness for both lightweight and high-capacity settings. In parallel, we investigate how performance scales with training data size, a key factor in offline distillation. Following GET [24], we train on 250K, 500K, and 1M samples and report FID scores on unconditional CIFAR-10 in Fig. 5B. Results show that more data yields better generation quality. For instance, increasing the dataset from 250K to 1M samples results in a $\approx 33\%$ FID reduction, showing the importance of data scale. While our method remains data-efficient, additional data can enhance performance.

**Complexity analysis.** Offline distillation can offer advantages over online methods, both memory-wise and time-wise (see App. E.5). KDM further improves over GET. As shown in Tab. 4, it matches GET in parameter count but achieves a $4\times$ speedup per training iteration and over $8\times$ faster sampling. Notably, KDM-F (App. C) maintains this efficiency even with spectral regularization. Specifically, we evaluate KDM and KDM-F with an additional eigen-loss (EL) term, observing that KDM-F + EL requires training time comparable to KDM (without EL), whereas KDM + EL incurs a longer training duration. These results underscore the practical scalability of our approach.

# 7    Discussion

In this work, we present two key observations. First, we reinterpret the distillation process through the lens of dynamical systems theory. Second, we uncover the semantic structure in the mapping from latent space to data space in trained diffusion dynamics. Motivated by these insights, we propose a Koopman-based framework for distilling diffusion models, bridging

Table 4: Parameters in millions, time in seconds. RTX 4090 GPU used with batch size 128.

| Metric | GET | KDM | KDM + EL | KDM-F + EL |
|---|---|---|---|---|
| Params (M) | 62 | 62 | 62 | 65 |
| Train (s) | 3.60 | 0.91 | 4.82 | 0.97 |
| Sampling (s) | 937 | 114 | 114 | 116 |

operator theory and generative modeling. We theoretically establish the existence of a Koopman operator that captures the underlying generative process, and demonstrate, both theoretically and empirically, that our method preserves its structure. This perspective also offers insights into potential failure modes. Beyond its conceptual contributions, our approach is practical and efficient, achieving highly competitive single-step offline distillation. We further analyze its robustness and scalability. We hope this work lays the foundation for future directions, including leveraging richer training signals and extending distillation techniques to advanced text-to-image generative models. See App. F for further discussion.

# Acknowledgments

This research was partially supported by the Lynn and William Frankel Center of the Computer Science Department, Ben-Gurion University of the Negev, an ISF grant 668/21, an ISF equipment grant, and by the Israeli Council for Higher Education (CHE) via the Data Science Research Center, Ben-Gurion University of the Negev, Israel.

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

# A Proof of Theorem 5.1

For the reader's convenience, we restate the theorem.

**Theorem** (Finite Koopman Approximation for Analytic Dynamics). *Let $\Phi : \mathbb{R}^n \to \mathbb{R}^n$ be an analytic map, and let $x_T \sim \mathcal{N}(0, I_n)$. Then, for any $\epsilon > 0$, there exists a linear operator $C \in \mathbb{R}^{d \times d}$ such that*

$$\mathbb{E}\left[\|\xi(\Phi(x_T)) - C\xi(x_T)\|^2\right] \le \epsilon \,,$$

*where $\xi : \mathbb{R}^n \to \mathbb{R}^d$ denotes the observable lifting map. Moreover, the required dimension $d$ grows at most polynomially with $1/\epsilon$, and the mapping $\Phi$ can be approximated arbitrarily well using multivariate polynomials up to degree $d$.*

*Proof.* Since $\Phi : \mathbb{R}^n \to \mathbb{R}^n$ is analytic, for every $\delta > 0$ and every compact set $K \subset \mathbb{R}^n$, there exists a multivariate polynomial map $p : \mathbb{R}^n \to \mathbb{R}^n$ such that

$$\sup_{x \in K} \|\Phi(x) - p(x)\| \le \delta \,,$$

which follows from the theory of approximation by Taylor polynomials for analytic functions [81].

Let $x_T \sim \mathcal{N}(0, I_n)$. Since $x_T$ has exponentially decaying tails [83], for any $\eta > 0$ there exists $R > 0$ such that

$$\mathbb{P}(x_T \notin B_R) \le \eta,$$

where $B_R = \{x \in \mathbb{R}^n : \|x\| \le R\}$. Restricting attention to $B_R$, we consider the polynomial approximation $p$ of $\Phi$ with error at most $\delta$.

Now, following the result of Iacob et al. [33], for the polynomial map $p$, there exists a finite-dimensional lifting map $\xi : \mathbb{R}^n \to \mathbb{R}^d$ consisting of monomials up to some degree, and a linear operator $C \in \mathbb{R}^{d \times d}$, such that

$$\xi(p(x)) = C\xi(x)$$

for all $x \in \mathbb{R}^n$. In particular, on $B_R$,

$$\|\xi(\Phi(x)) - C\xi(x)\| \le \|\xi(\Phi(x)) - \xi(p(x))\|,$$

since $\xi(p(x)) = C\xi(x)$ exactly.

Since $\xi$ is a smooth (polynomial) map, there exists a Lipschitz constant $L > 0$ depending on the derivatives of $\xi$ over $B_R$, such that

$$\|\xi(\Phi(x)) - \xi(p(x))\| \le L\|\Phi(x) - p(x)\| \le L\delta$$

for all $x \in B_R$.

Thus, for $x_T \in B_R$,

$$\|\xi(\Phi(x_T)) - C\xi(x_T)\| \le L\delta.$$

Outside $B_R$, the trivial bound

$$\|\xi(\Phi(x_T)) - C\xi(x_T)\| \le 2M$$

holds for some finite constant $M > 0$, due to the polynomial growth of $\xi$ and the Gaussian decay of $x_T$.

Hence, the expected Koopman error satisfies

$$\mathbb{E}\left[\|\xi(\Phi(x_T)) - C\xi(x_T)\|^2\right] \le (1 - \eta)L^2\delta^2 + \eta(2M)^2.$$

Given any $\epsilon > 0$, we can choose $\eta$ sufficiently small so that $\eta(2M)^2 \le \epsilon/2$, and then $\delta$ sufficiently small so that $(1 - \eta)L^2\delta^2 \le \epsilon/2$.

Notably, the degree of the polynomial map $p$ needed, and hence the number of monomials $d$ in $\xi$, grows at most polynomially with $1/\delta$ and thus with $1/\epsilon$, concluding the proof.

We further note that the function $\Phi$ implemented by `DhariwalUNet` [35] is analytic. Specifically, each module composing the network—including convolutions, SiLU activations, residual connections, attention mechanisms, and group normalization with $\epsilon > 0$—is an analytic function. Since compositions of analytic functions remain analytic, it follows that the entire forward pass $\Phi$ is analytic on $\mathbb{R}^n$.

$\square$

# B  Proof of Theorem 5.2

We restate the theorem for the readers' convenience.

**Theorem** (Semantic Proximity via Koopman-Invariant Coordinates). *Let $\Phi_t : \mathbb{R}^n \to \mathbb{R}^n$ be the reverse diffusion flow of a trained model (from time $T$ to $0$), and let $\mathcal{O} \subset L^2(\mathbb{R}^n)$ be a finite-dimensional subspace such that:*

1. *$\mathcal{K}_t \mathcal{O} \subset \mathcal{O}$ for all $t \in [0, T]$,*

2. *$\exists\, C_t : \mathbb{R}^d \to \mathbb{R}^d$ such that $\mathcal{K}_t \xi = \xi \circ \Phi_t = C_t \xi$ for all $\xi \in \mathcal{O}$,*

3. *$\Phi_T$ is locally Lipschitz.*

*For any $x_T^1, x_T^2 \in \mathbb{R}^n$, define $x_0^j := \Phi_T(x_T^j)$ for $j = 1, 2$, and let $\xi : \mathbb{R}^n \to \mathbb{R}^d$ be the vector-valued observable $\xi(x) := (\xi_1(x), \ldots, \xi_d(x)) \in \mathcal{O}$. Then:*

$$\|x_0^1 - x_0^2\| \le L \cdot \|\Phi_T(x_T^1) - \Phi_T(x_T^2)\| \le L \cdot \|C_T\| \cdot \|\xi(x_T^1) - \xi(x_T^2)\|.$$

*Proof.* Let $\Phi_t : \mathbb{R}^n \to \mathbb{R}^n$ denote the reverse diffusion dynamics. Based on the proof of Thm. 5.1, there exists a linear operator $C_t : \mathbb{R}^d \to \mathbb{R}^d$ satisfying

$$\mathcal{K}_t \xi = \xi \circ \Phi_t = C_t \xi\,,$$

for every $\xi \in \mathcal{O}$ and $t \in [0, T]$.

Define the vector-valued map for the basis $\{\xi_j\}$ of $\mathcal{O}$

$$\xi(x) := (\xi_1(x), \ldots, \xi_d(x)) \in \mathbb{R}^d.$$

Since the $\xi_j$ are smooth and capture independent semantic directions, $\xi$ is locally bi-Lipschitz near typical $x_T$. That is, there exist constants $C_1, C_2 > 0$ such that

$$C_1 \|\xi(x_T^1) - \xi(x_T^2)\| \le \|x_T^1 - x_T^2\| \le C_2 \|\xi(x_T^1) - \xi(x_T^2)\|.$$

Because $\Phi_T$ is locally Lipschitz, there exists $L > 0$ such that

$$\|\Phi_T(x_T^1) - \Phi_T(x_T^2)\| \le L \|x_T^1 - x_T^2\|.$$

Substituting the bi-Lipschitz inequality yields

$$\|\Phi_T(x_T^1) - \Phi_T(x_T^2)\| \le L C_2 \|\xi(x_T^1) - \xi(x_T^2)\|.$$

Moreover, by Koopman linearity,

$$\xi(\Phi_T(x_T^j)) = C_T \xi(x_T^j), \quad j = 1, 2,$$

and therefore

$$\xi(\Phi_T(x_T^1)) - \xi(\Phi_T(x_T^2)) = C_T(\xi(x_T^1) - \xi(x_T^2)).$$

Thus, proximity in $\xi(x_T)$ coordinates implies proximity of final samples $x_0^1$ and $x_0^2$ under $\Phi_T$.

$\square$

## C  Factorized Koopman Matrix for Fast Eigenspecturm penalty

To enable control over eigenvalues of the Koopman operator, we describe an efficient implementation of our Koopman-based Distillation Method (KDM), which we term KDM-F, alleviating the need for computation of the eigendecomposition during training. Since every matrix $C := C_\eta$ is diagonalizable up to an arbitrarily small perturbation of the entries [2], we can write:

$$C = P\Lambda P^{-1}$$

where $P \in \mathbb{C}^{d\times d}$ is an invertible matrix and $\Lambda = \mathrm{diag}(\lambda_1, \ldots, \lambda_d) \in \mathbb{C}^{d\times d}$. This way enables efficient implementation of spectral penalties on $\Lambda$, without the need to compute it on the fly during training.

To represent this decomposition within a real-valued neural network framework, we learn the real and imaginary components of the Koopman eigenvectors separately. Specifically, two orthonormal matrices, $P_{\mathrm{re}}$ and $P_{\mathrm{im}}$, represent the real and imaginary parts of the eigenvectors. These are combined into a real-valued block matrix using the standard transformation for complex-to-real matrix representation:

$$P = \begin{bmatrix} P_{\mathrm{re}} & -P_{\mathrm{im}} \\ P_{\mathrm{im}} & P_{\mathrm{re}} \end{bmatrix} \in \mathbb{R}^{2d\times 2d} .$$

Rather than explicitly computing the inverse $P^{-1}$, we parameterize a separate learnable matrix $P_{\mathrm{inv}}$ with the same structure, constructed analogously from two unconstrained real matrices. This improves flexibility and avoids numerical instability associated with matrix inversion.

The eigenvalues of the Koopman operator are encoded using polar coordinates. Specifically, the modulus and phase of each eigenvalue are parameterized as

$$\lambda_j = e^{-\exp(\nu_j)} \cdot e^{i\theta_j} = e^{-\exp(\nu_j)} \left(\cos(\theta_j) + i\sin(\theta_j)\right),$$

where $\nu_j$ and $\theta_j$ are learnable parameters. Let $\lambda_{\mathrm{re}} = (\mathrm{real}(\lambda_j)), \lambda_{\mathrm{im}} = (\mathrm{imag}(\lambda_j)) \in \mathbb{R}^d$ be the aggregated vector of real and imaginary parts of the eigenvalues, respectively. We assemble these values into a real-valued block-diagonal matrix:

$$\Lambda = \begin{bmatrix} \mathrm{diag}(\lambda_{\mathrm{re}}) & -\mathrm{diag}(\lambda_{\mathrm{im}}) \\ \mathrm{diag}(\lambda_{\mathrm{im}}) & \mathrm{diag}(\lambda_{\mathrm{re}}) \end{bmatrix} \in \mathbb{R}^{2d\times 2d} .$$

At inference time, the latent state vector $z \in \mathbb{R}^d$ is first augmented to $\mathbb{R}^{2d}$ by concatenating a zero vector: $\tilde{z} = [z; 0]$. The evolution under the Koopman operator is then computed as

$$z_{\mathrm{next}} = P_{\mathrm{inv}}\Lambda P\tilde{z} .$$

Finally, we discard the imaginary part and retain only the first $d$ entries to return a real-valued output of the same dimensionality as the input.

This formulation enables us to simulate rich, expressive dynamics through a fully linear evolution in the lifted Koopman space, while maintaining computational tractability and interpretability. Moreover, this decomposition is particularly amenable to distillation: the structure of the transformation ensures that semantic and temporal coherence is preserved during student training, as the model explicitly disentangles dynamics into magnitude and phase components.

# D Additional Experiments and Analysis

## D.1 Ablation Studies

**Loss Ablation Study.** We conduct an ablation study to evaluate the contribution of each loss component. As shown in Tab. 5, using only $\mathcal{L}_{\text{pred}}$ or any pairwise combination of the Koopman-related losses still enables the model to learn, as indicated by moderate FID and IS scores. However, incorporating all three losses together, including $\mathcal{L}_{\text{Koopman}}$, creates a synergistic effect—reducing the FID from around 10 to 7.83 and achieving a higher IS than any partial combination. Finally, we highlight the importance of incorporating an adversarial setup, which further improves both FID and IS scores by enhancing perceptual quality.

Note, integrating adversarial losses into alternative distillation methods for diffusion models is often non-trivial and presents significant challenges. Many existing approaches lack the architectural flexibility or training stability required to effectively incorporate such objectives, unlike our proposed KDM. Notably, we extended GET with adversarial training, which improved its FID from 6.91 to 6.13. However, KDM still achieves a superior FID of 4.68, demonstrating its effectiveness beyond prior methods.

Table 5: Koopman losses ablation; bold is best and underscore is second-best.

|  | $\mathcal{L}_{\text{lat}}$ | $\mathcal{L}_{\text{pred}}$ | $\mathcal{L}_{\text{rec}}$ | $\mathcal{L}_{\text{lat}} + \mathcal{L}_{\text{pred}}$ | $\mathcal{L}_{\text{rec}} + \mathcal{L}_{\text{pred}}$ | $\mathcal{L}_{\text{rec}} + \mathcal{L}_{\text{lat}}$ | $\mathcal{L}_{\text{Koopman}}$ | $\mathcal{L}$ | GET + Adv |
|---|---|---|---|---|---|---|---|---|---|
| FID | 487 | 11.2 | 457 | 9.53 | 10.7 | 11.16 | 7.83 | **4.97** | 6.13 |
| IS | 1.00 | 8.23 | 1.00 | 8.82 | 8.26 | 8.68 | 8.83 | **9.22** | 8.98 |

**Ablation on Loss Weighting and $\lambda_{\text{adv}}$ Sensitivity** We investigate the effect of different weightings for the components of the Koopman loss, namely $\mathcal{L}_{\text{rec}}$, $\mathcal{L}_{\text{lat}}$, and $\mathcal{L}_{\text{pred}}$, as well as the influence of the adversarial weight $\lambda_{\text{adv}}$. The results indicate that the model is robust to moderate changes in these loss weights, with FID scores remaining within a narrow range (4.94–5.03). For the adversarial term, we varied $\lambda_{\text{adv}} \in \{0.5, 1.0, 0.25\}$ and observed that $\lambda_{\text{adv}} = 1.0$ achieved the best trade-off between fidelity and stability, yielding an FID of 4.57 compared to 4.98 for $\lambda_{\text{adv}} = 0.5$ and 4.67 for the default $\lambda_{\text{adv}} = 0.25$ setting used in the main paper. These findings suggest that KDM remains stable under different loss scalings and that its performance is not overly sensitive to the choice of $\lambda_{\text{adv}}$, demonstrating strong robustness to training hyperparameters.

**Empirical Investigation of Koopman Matrix Size.** We evaluate the impact of varying the size of the Koopman matrix. As shown in Tab. 6, smaller matrices tend to generalize better than larger ones. This result is somewhat counterintuitive, as the scaling experiments in Sec. 6.2 demonstrate a clear trend where increasing the number of parameters typically improves performance. We hypothesize that smaller matrices generalize more effectively, and we leave a deeper understanding of this phenomenon—through the lens of dynamical systems theory and empirical analysis—for future research.

Table 6: FID scores across matrix sizes on the CIFAR-10 unconditional task; bold is best.

| Size | 1024 | 4096 | 9216 |
|---|---|---|---|
| FID | **5.08** | 7.24 | 9.41 |

## D.2 Training Stability and Adversarial Loss Convergence

To evaluate the training stability of *KDM*, we compare the generator and discriminator loss dynamics of KDM and GET across uniformly sampled training steps. Despite the inclusion of an adversarial component in KDM, both losses exhibit stable and convergent behavior throughout training, indicating that the adversarial term does not introduce significant instability in practice. Fig.6 illustrates the generator and discriminator loss trajectories for both KDM and GET. We discard the 0% step due to corrupted values in the raw logs. Overall, the results confirm that KDM maintains stable optimization behavior comparable to GET, while benefiting from enhanced perceptual quality.

Generator and Discriminator Loss Convergence for KDM and GET

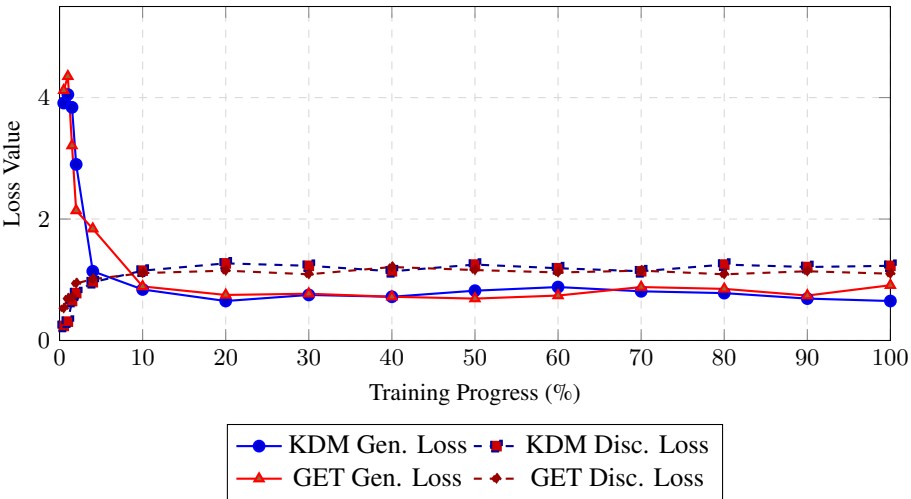

Figure 6: Training loss convergence comparison between KDM and GET. Despite including an adversarial loss, KDM maintains stable generator and discriminator convergence comparable to GET.

## D.3 Additional Qualitative Results

We include additional qualitative results of our method, extending the evaluation presented in the main text on more datasets and testing scenarios.

**Additional Comparison of Student vs Teacher.** In Figs. 7, 8, 9, and 10, we present generations from our model alongside those of the original EDM model. Corresponding blocks indicate samples generated from the same initial noise.

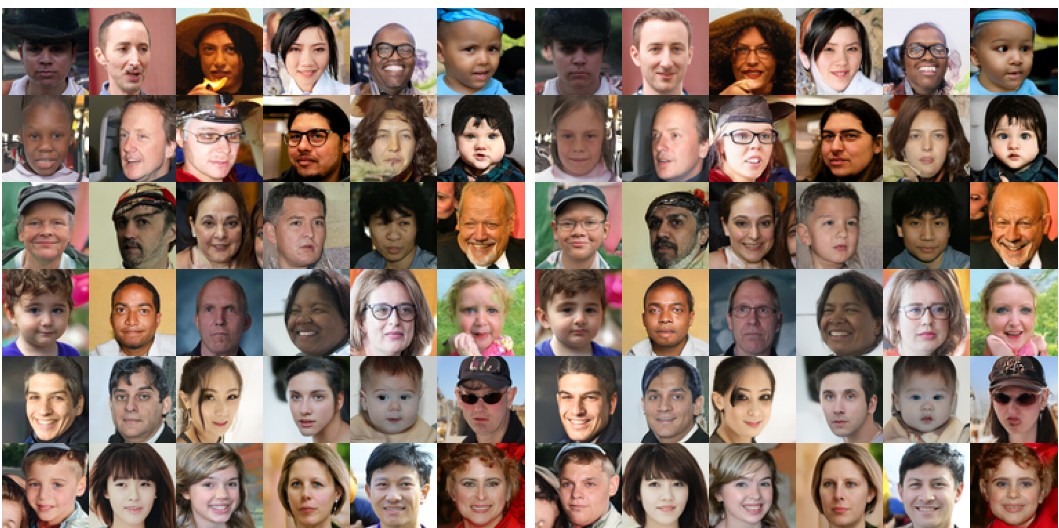

Figure 7: Unconditional Generation of FFHQ. Sampled images using 79 NFEs with EDM [35] (left), and using 1 NFE with KDM (right).

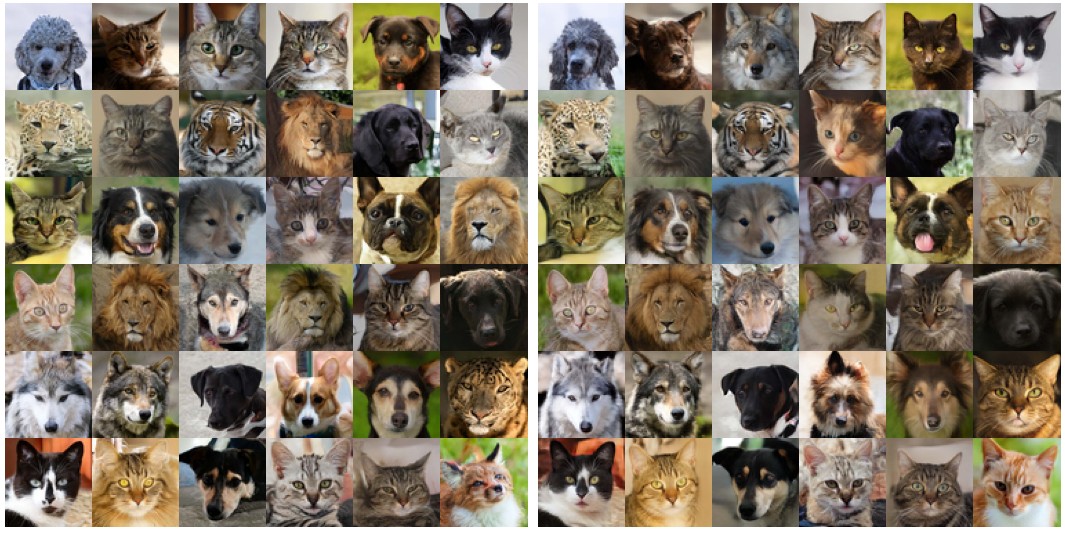

Figure 8: Unconditional Generation of FFHQ. Sampled images using 79 NFEs with EDM [35] (left), and using 1 NFE with KDM (right).

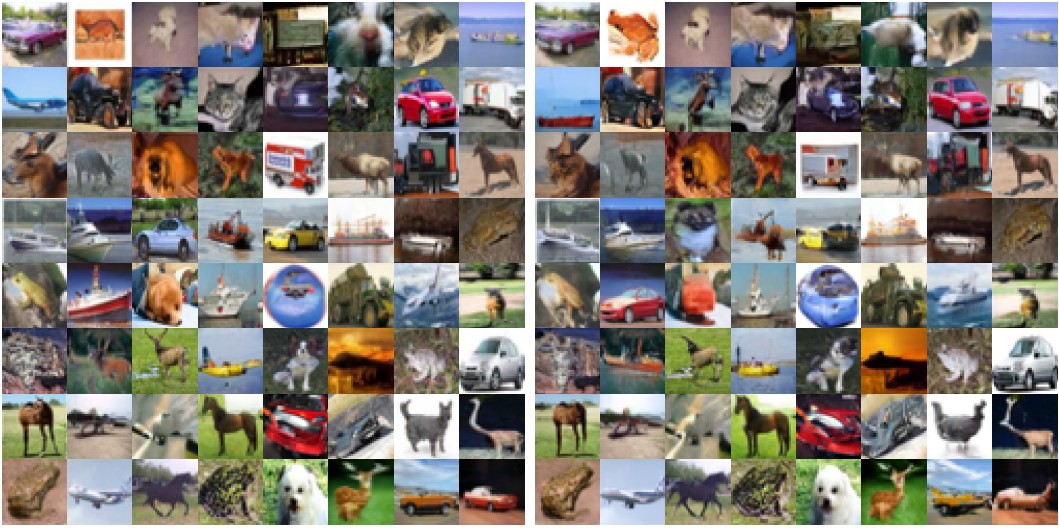

Figure 9: Unconditional Generation of CIFAR-10. Sampled images using 35 NFEs with EDM [35] (left), and using 1 NFE with KDM (right).

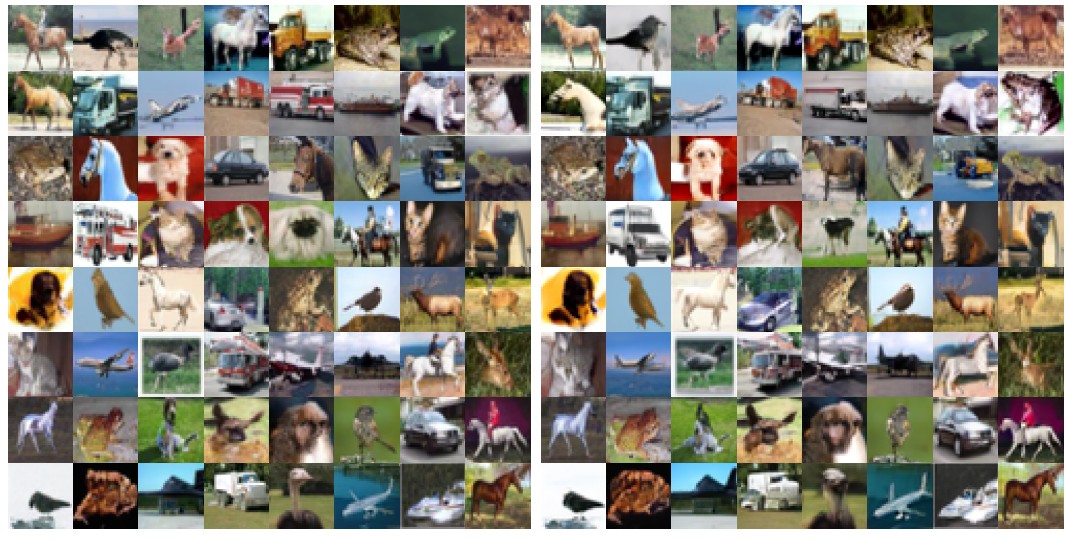

Figure 10: Conditional Generation of CIFAR-10. Sampled images using 35 NFEs with EDM [35] (left), and using 1 NFE with KDM (right).

**Additional Local Structure Analysis.** We extend the experiment from Sec. 6.1 to AFHQv2, conditional CIFAR-10, and unconditional CIFAR-10 datasets, as shown in Fig. 11, Fig. 12, and Fig. 13, respectively. The results reinforce the conclusions of the main experiment: both EDM and our distilled model exhibit local semantic coherence, preserving features such as position, color, identity, and more—even when traversing farther from the original point in noise space.

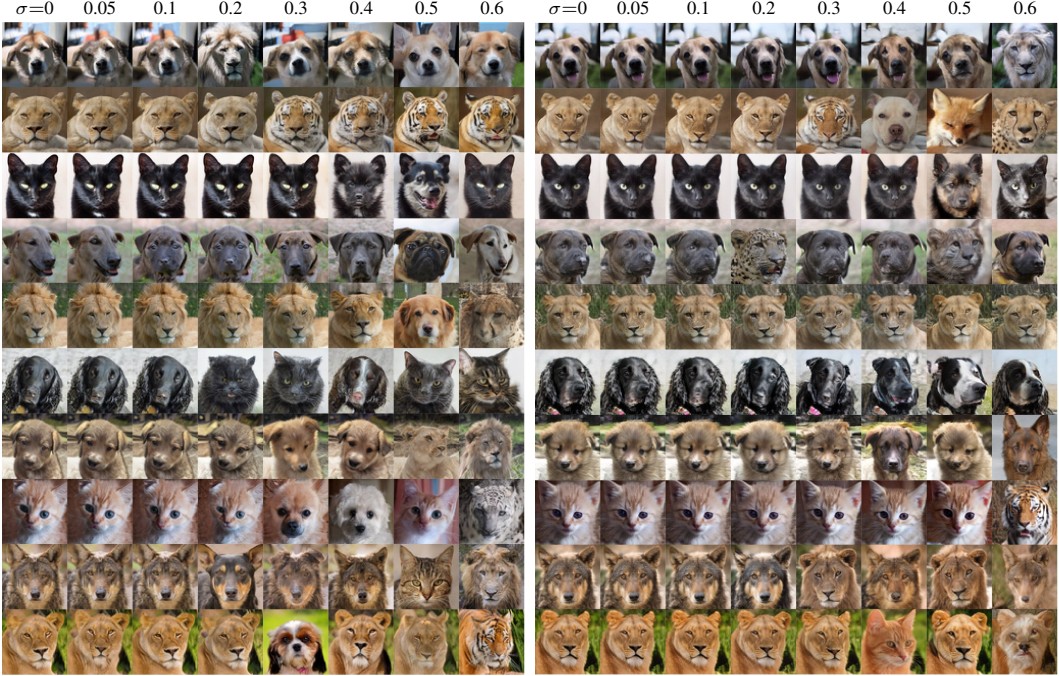

Figure 11: Unconditional AFHQv2: Comparison of neighborhood exploration across different noise levels $\sigma$. Left: EDM [35]. Right: our method. Each image corresponds to a small perturbation around the original noise sample.

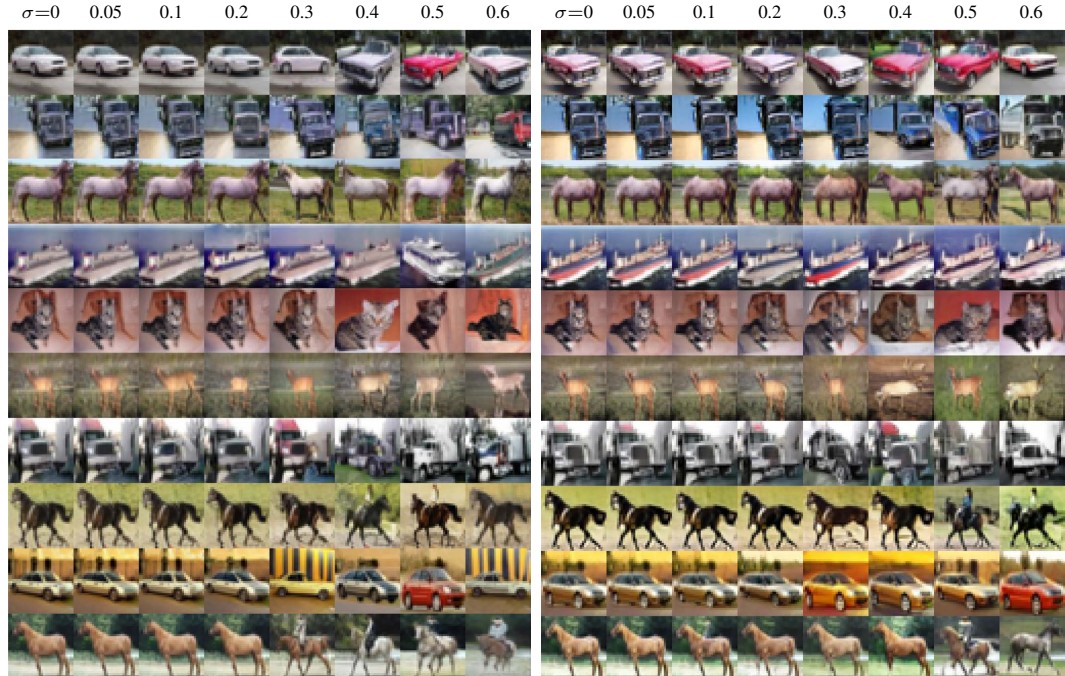

Figure 12: Conditional CIFAR-10: Comparison of neighborhood exploration across different noise levels $\sigma$. Left: EDM [35]. Right: our method. Each image corresponds to a small perturbation around the original noise sample.

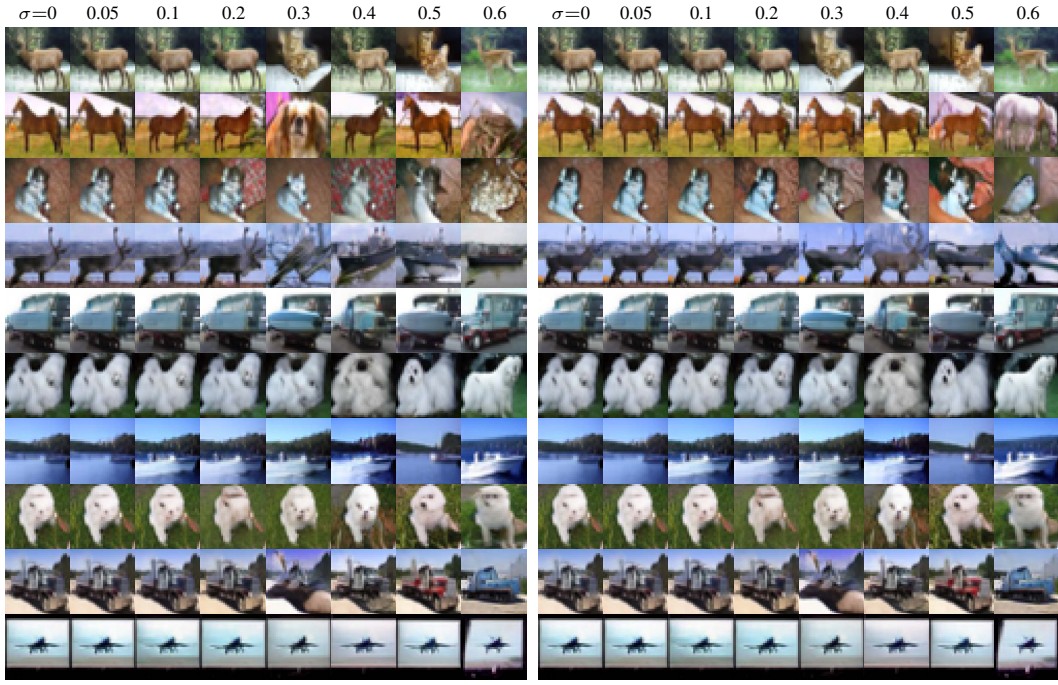

Figure 13: Unconditional CIFAR-10: Comparison of neighborhood exploration across different noise levels $\sigma$. Left: EDM [35]. Right: our method. Each image corresponds to a small perturbation around the original noise sample.

# E  Additional Details

## E.1  Training and Sampling Procedures Psudo-codes

The training process of the model follows a Koopman-inspired backward mapping approach, augmented with adversarial supervision. Given a clean-noisy pair $(x_0, x_T)$, the model learns a latent Koopman operator that evolves the noisy encoding backward to approximate the clean encoding, while a decoder reconstructs the input from the latent space. The model also supports class-conditional dynamics via Koopman control.

---

**Algorithm 1** Training Procedure

---

**Require:** Training pairs $(x_0, x_T)$, labels $y$ (optional), model $E_\phi$, $E_\psi$, $C_\eta$, $D_\psi$ discriminator $D_\gamma$
1: **for** each training step **do**
2:     Encode $z_0 \leftarrow E_\phi(x_0, 0, y)$
3:     Encode $z_T \leftarrow E_\varphi(x_T, T, y)$
4:     Add noise: $\tilde{z}_0 \leftarrow z_0 + \epsilon_0$, $\tilde{z}_T \leftarrow z_T + \epsilon_T$, where $\epsilon_T, \epsilon_0 \sim \mathcal{N}(0, 0.4)$
5:     Push via Koopman: $z_0^{\text{push}} \leftarrow C_\eta(\tilde{z}_T)$
6:     **if** conditional Koopman **then**
7:         Add control: $z_0^{\text{push}} \leftarrow z_0^{\text{push}} + C_\mu(y)$
8:     **end if**
9:     Decode: $\hat{x}_0 \leftarrow D_\psi(\tilde{z}_0, 0, y)$
10:    Decode pushed: $\hat{x}_0^{\text{push}} \leftarrow D_\psi(z_0^{\text{push}}, 0, y)$
11:    Compute latent loss: $\mathcal{L}_{\text{lat}} \leftarrow \|z_0 - z_0^{\text{push}}\|^2$
12:    Compute reconstruction loss: $\mathcal{L}_{\text{rec}} \leftarrow d(x_0, \hat{x}_0)$
13:    Compute full pass loss: $\mathcal{L}_{\text{pred}} \leftarrow d(x_0, \hat{x}_0^{\text{push}})$
14:    Compute model adversarial loss: $\mathcal{L}_{\text{adv}}^G \leftarrow CE(1, D_\gamma(\hat{x}_0^{\text{push}}))$
15:    Compute discriminator loss: $\mathcal{L}_{\text{adv}}^D \leftarrow CE(1, D_\gamma(x_0)) + CE(0, D_\gamma(\hat{x}_0^{\text{push}}))$
16:    model Combine losses:

$$\mathcal{L}_{\text{total}} \leftarrow \mathcal{L}_{\text{rec}} + \mathcal{L}_{\text{pred}} + \mathcal{L}_{\text{lat}} + w_{\text{adv}} \cdot \mathcal{L}_{\text{adv}}^G$$

17:    Update $E_\phi$, $E_\varphi$, $C_\eta$, $C_\mu$, $D_\psi$ and $D_\gamma$ via backpropagation
18: **end for**

---

$d$ is a distance measure such as MSE or LPIPS. We utilize LPIPS. $CE$ is cross-entropy loss. At inference time, the model can sample clean images by reversing from random Gaussian noise $x_T$, applying the Koopman operator in latent space, and decoding to the image space.

---

**Algorithm 2** Sampling Procedure

---

**Require:** Koopman model: $E_\phi$, $C_\eta$, $C_\mu$, $D_\psi$, labels $y$ (optional)
1: Sample noise $x_T \sim \mathcal{N}(0, \sigma^2)$
2: Encode: $z_T \leftarrow E_\varphi(x_T, T, y)$
3: Push: $z_0^{\text{push}} \leftarrow C_\eta(z_T)$
4: **if** conditional Koopman **then**
5:     Add control: $z_0^{\text{push}} \leftarrow z_0^{\text{push}} + C_\mu(y)$
6: **end if**
7: Decode: $\hat{x}_0 \leftarrow D_\psi(z_0^{\text{push}}, 0, y)$
8: **return** Clean sample $\hat{x}_0$

---

## E.2  Datasets Collection

For the **Checkerboard** dataset, we sampled using 10 NFEs with EDM [35] or flow matching [46]. Following [24], we sampled **CIFAR-10** with 35 NFEs from EDM. For **FFHQ** and **AFHQv2**, we used 79 NFEs using EDM, as recommended.

### E.3 Method Implementation Details

We provide the full code implementation for the Checkerboard and CIFAR-10 (both conditional and unconditional) datasets in the supplementary material. Due to memory constraints, the implementations for FFHQ and AFHQv2 will be released shortly. In this section, we briefly describe the implementation details of each method, while we encourage the reader to refer to the codebase for exact configurations. We train our models for 800k iterations with the Adam optimizer [39] at a fixed learning rate of 3e-4. We do not use warm-up, weight decay, or any learning-rate schedule. All runs are on a single NVIDIA A6000 GPU or RTX4090. Note that our method can fully support distributed training.

**Checkerboard.** For this dataset, we build upon the implementation of [47]. Both $x_0$ and $x_T$ are encoded using the same denoising architecture as described in their work. The decoding of $z_0$ is performed using the inverse of this module. The Koopman operator is implemented as a simple linear layer. Complete implementation details are available in the shared code.

**CIFAR-10, FFHQ, and AFHQv2.** We adopt the SongUNet architecture from [35] for encoding $x_0$ and $x_T$, as well as for decoding $z_0$. To reduce latent dimensionality, the output channels of the network are set to 1, effectively reducing the latent space size by a factor of 3. The Koopman operator is implemented either as a standard linear transformation or using our custom Fast Koopman module. To maintain architectural fairness, we downscale all networks to match the total parameter count of the original Karras network [35] and the GET distillation model [24]. The FFHQ and AFHQv2 configurations follow a similar encoder-decoder structure with one key difference: a linear projection layer is appended to the end of the encoder and the beginning of the decoder to further control the latent space dimensionality. All other aspects remain consistent across datasets. Note that the discriminator is trained using a separate optimizer, akin to the Koopman module's optimizer.

**Hyperparameters and Training.** We provide complete training scripts along with plug-and-play configuration files in the supplementary material. An overview of the key hyperparameters used for each dataset is summarized in Tab. 7. Overall, our method requires minimal hyperparameter tuning and we did not perform any hyperparameter search.

- **Unet Out Channels**: Controls the number of output channels in SongUNet.

- **Unet Model Channels**: Determines the internal channel width, affecting the model's capacity.

- **Noisy Latent Injection**: A Gaussian noise term $\epsilon \sim \mathcal{N}(0, 1)$ added to latents (as described in the pseudocode) to improve generalization and mitigate numerical instabilities.

- **Adversarial Weight**: Applied to the adversarial loss term in our generator loss; not used for the discriminator loss directly.

- **Linear Module**: Indicates the presence of an additional linear layer for compressing the latent space, applied at the output of the encoder and the input of the decoder.

Table 7: Model Hyperparameters Across Datasets

| Configuration | Checkerboard | CIFAR-10 | FFHQ | AFHQv2 |
|---|---|---|---|---|
| Batch Size | 4096 | 128 | 64 | 64 |
| Learning Rate (LR) | 0.0003 | 0.0003 | 0.0003 | 0.0003 |
| Iterations | 800k | 800k | 800k | 800k |
| Unet Out Channels | 1 | 1 | 1 | 1 |
| Unet Model Channels | 64 | 64 | 32 | 32 |
| Noisy Latent Injection | 0.4 | 0.4 | 0.4 | 0.4 |
| Adversarial Weight | 0.01 | 0.01 | 0.01 | 0.01 |
| Linear Module | No | No | Yes | Yes |
| Latent Dim Size | 1024 | 1024 | 512 | 512 |

Table 8: Human Evaluation of Image Realism. The reported ratio reflects the number of times participants selected the model-generated collage as more realistic, out of a total of 300 evaluations.

|  | EDM (Teacher) | KDM (Student) |
|---|---|---|
| Ratio | 45% | 55% |

## E.4 Human Evaluation Experiment

While the Fréchet Inception Distance (FID) is widely used to evaluate the quality of images generated by generative models, recent studies have highlighted its limitations in aligning with human judgments of image realism. For instance, Jayasumana et al. [34] demonstrate that FID can contradict human evaluations, particularly in assessing perceptual quality and semantic alignment, due to its reliance on Inception-v3 features and the assumption of Gaussian distributions. Similarly, Otani et al. [64] emphasize that automatic metrics like FID often fail to capture the nuanced aspects of human perception, underscoring the need for standardized human evaluation protocols. Furthermore, Kynkäänniemi et al. [44] reveal that FID can be manipulated by aligning top-N class histograms without genuinely improving image quality, indicating potential biases in the metric. These findings suggest that while FID provides a useful quantitative signal, it may not fully encapsulate human perceptions of image realism, highlighting the importance of incorporating human evaluations when assessing generative models.

Therefore, in the main paper, we report the results of a human evaluation experiment designed to compare our method against EDM. Specifically, we curated 10 image collages (6×6) of human faces generated by the teacher model (EDM, 79 NFEs) and 10 collages generated by our student model (1 NFE), using samples from the FFHQ dataset. Each comparison slide presents one collage from each model, displayed side-by-side as illustrated in Fig. 4, with the order randomized. We then asked 30 human raters to evaluate 10 such slides and choose, for each, which collage appeared more realistic. If both appeared equally realistic, they were instructed to make a random choice. This process yielded 300 total decisions. We report in the main paper and in Tab. 8, the proportion of times each model was preferred. Finally, we attach to the appendix both the slides that we sent to the practitioners and the Excel that aggregates the results. The results indicate that decisions were made almost at random, suggesting that participants had difficulty distinguishing between the student and teacher generations. This implies that, despite existing FID gaps, human evaluators found it hard to perceive meaningful differences. Participants reported that both collages included highly realistic images, as well as images with noticeable artifacts that appeared unrealistic, in their judgment.

## E.5 Cost and Efficiency of Offline Distillation

An important consideration in offline distillation is the total computational cost, which includes both the training of the student model and the one-time cost of generating the teacher-produced noise–data pairs. While this pair generation step incurs an initial overhead, it is performed only once and can be reused across multiple student architectures or experiments, effectively amortizing its cost. Following the analysis of [24], which explicitly incorporates the cost of data-pair generation into its complexity assessment, the overall computational load of offline distillation remains favorable compared to online approaches. This is primarily because online distillation requires repeated access to the full teacher model throughout training, resulting in substantially higher neural function evaluations and memory usage. In our experiments on the main paper, we observe that even when accounting for the data generation step, our method achieves a markedly better trade-off between cost and performance than prior offline frameworks such as GET. These results collectively support that, despite the initial preprocessing overhead, offline distillation constitutes an efficient and scalable alternative for practical deployment. To see the full analysis between online and offline approaches, please refer to [24].

## E.6 Parameter Scaling

We report the full results of the scaling experiment shown in Sec . 6.2, including IS score from the parameter scaling comparison in Tab. 9. The results demonstrate that our approach consistently outperforms GET in terms of FID across all model sizes, except for IS in 20M, indicating superior perceptual quality. Notably, the improvement is more pronounced as the model size increases, with

our 84M model achieving an FID of 4.89 compared to 7.19 for GET. In terms of IS, our method matches or surpasses GET, with particularly strong results at larger scales. These results underscore the scalability and effectiveness of our approach.

Table 9: FID and IS scores for different model sizes, comparing our approach with GET.

|  | 20M | | 38M | | 84M | |
|  | Ours | GET | Ours | GET | Ours | GET |
|---|---|---|---|---|---|---|
| FID | **10.2** | 10.72 | **5.04** | 8.00 | **4.89** | 7.19 |
| IS | 8.50 | **8.69** | **9.12** | 9.03 | **9.11** | 9.09 |

### E.7    Extended Outlier Analysis

In Sec. 6.1, we present the results of our model's component (left side) in the experiment shown in Fig. 14. Outlier detection was performed using the DBSCAN algorithm with a distance threshold of 0.15. Interestingly, our model exhibits a slightly increased number of outliers. We hypothesize that this may be due to early stopping during training, as our primary objective was not to maximize performance on this particular dataset, but rather to gain insight and intuition into the nature of the learned dynamics. Finally, we observe that the outlier phenomenon is inherited from the teacher model (right side), suggesting that it is a more general characteristic of the generative dynamics rather than an artifact introduced by the distillation process.

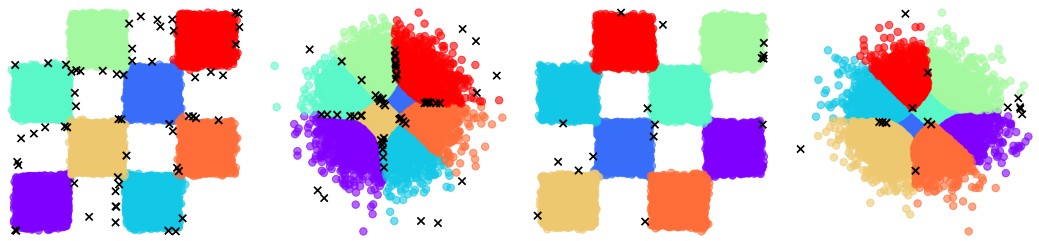

Figure 14: Outlier analysis on generated checkerboard patterns. The two left images are our model (student), and the two images on the right are from the flow matching model (teacher).

## F    Additional Discussion of Future Work and Current Limitations

In this study, we took a significant step toward leveraging dynamical systems theory for offline distillation, demonstrating its strong potential in this domain. In this section, we outline several future research directions that we believe are both promising and intellectually compelling.

**Standalone Koopman Generative Model.**    Koopman theory enables modeling nonlinear dynamical systems in a transformed space where they can be analyzed linearly. This raises an intriguing question: *Can the Koopman framework serve as a standalone generative model, capable of learning data distributions from scratch?* While the question remains open, exploring this direction could yield a novel and theoretically grounded framework for generative modeling.

**Utilization of Full Trajectory Information.**    Our current offline setup relies solely on noise-data pairs extracted from pre-trained teacher models. However, the diffusion process inherently includes intermediate steps that are discarded. We hypothesize that leveraging this full trajectory information could enhance trajectory alignment and improve student performance. Moreover, Koopman theory naturally accommodates multi-step signals, making this a promising direction for further investigation. Additionally, the current framework does not support a quality-versus-sampling-steps trade-off, which some online methods do enable. Future work incorporating the full diffusion trajectory could integrate this flexibility into our approach.

**Incorporating Broader Koopman Theory Literature.** The connection between Koopman theory and diffusion model distillation opens the door to a rich body of theoretical and practical tools. On the theoretical side, works on sample complexity and error bounds for Koopman operators [66, 14, 57] may provide insights into convergence guarantees. Information-theoretic perspectives [48] could offer new modeling constraints or regularization schemes. On the practical side, methods for rare-event sampling [87] may improve long-tail generation capabilities. Finally, while the current mathematical formulation supports discrete Koopman evaluation, the underlying theory naturally extends to continuous settings as well. Exploring these directions could substantially extend and generalize the impact of our framework.

**Enhancing the Adversarial Setup.** While adversarial components have been shown to be critical in many distillation methods [85, 38], our current implementation uses a basic discriminator architecture with standard loss functions and training schemes. We believe that integrating more advanced adversarial paradigms could significantly boost performance, potentially closing the gap with state-of-the-art online distillation methods.

**Decision Boundary Insights.** In our synthetic experiments, we identified a common cause underlying failure modes or "outliers" in the generative dynamics. Further investigation is needed to determine whether this phenomenon generalizes to more complex datasets. If it does, it could guide the development of strategies such as uncertainty estimation, selective sampling to avoid generating low-quality outputs, incorporating structural constraints into the latent space to prevent such failures, or introducing auxiliary self-supervised objectives—such as contrastive learning—to enhance the model's ability to distinguish between semantically similar groups.

**Toward Broader Applications.** Due to limited computational resources, we were unable to extend our offline distillation framework to more complex tasks, such as text-to-image generation, that require large-scale compute infrastructures. It will be interesting to explore how our method can scale or adapt to other domains and tasks. Additionally, because our method makes no assumptions about the initial and terminal states of the underlying dynamical system, we believe it is applicable to a broader class of generative models, including DDBMs [92], stochastic interpolants [1], and others [17].

