# OpenReview forum: "One-Step Offline Distillation of Diffusion-based Models  via Koopman Modeling"
_NeurIPS.cc/2025/Conference — NeurIPS 2025 poster_

### Official Review · Reviewer_zaaq · 2025-06-28

**Clarity:** 3
**Significance:** 2
**Originality:** 3
**Rating:** 3
**Confidence:** 5

**Summary:**

This paper presents Koopman Distillation, an approach rooted in Koopman theory for nonlinear dynamics, in which a pretrained diffusion model is distilled by learning an autoencoder on the latent representation of the model and a linear operator inside that space in order to transfer data to noise.

**Questions:**

N/A

**Ethical Concerns:**

["NO or VERY MINOR ethics concerns only"]

**Final Justification:**

I am reasonably confident my score of weak rejection (and have adjusted my confidence accordingly):
- The authors' main claims of success seem to hinge on what I see as an overly restrictive view of what constitutes "offline" distillation, as   they claim that access to *any* of the weights of the teacher model during distillation (including using part of the teacher as initialization for a discriminator as is done in LADD, Seaweed-APT, and UFO-Gen), even if there is no explicit supervision from the teacher as a diffusion model, would constitute as an online method. I have never seen this definition used before, and does not seem to be in line with the growing space of other offline methods. Additionally, upon further discussion, the author's stated that even initializing the student with the teacher would count as "online" distillation, despite this being the standard practice among nearly all modern distillation methods (both offline and online).
- It is clear to me that 2-rf++ actually is an offline method, thus showing that KDMs are not even SOTA for offline methods.
- Thus, despite the fascinating and concerted theoretical results throughout the paper, the experimental results are not strong enough to warrant acceptance.
- I think there is a bit of disconnect between the current structure of the paper and the practical results of their experiments. Throughout the rebuttal period, the authors highlighted the benefits of their definition of "offline" distillation from a privacy perspective, which I think are valid points. However, the paper barely focuses on this aspect (besides passing mentions of offline distillation in a broad sense) and instead is formatted much like a standard diffusion distillation paper, which does not help their relatively moderate experimental results. While I don't think this paper is currently ready for publication, I do think that a future version of this paper, which could empirically motivate *how* KDMs can be better than their defined "online" methods from a privacy perspective and in general focuses on KDMs' privacy preserving abilities, would make for a strong contribution.

**Limitations:**

The authors do recognize the limitations of their performance, though I do think the overall poor performance is a bit understated and downplayed.

**Quality:**

2

**Strengths And Weaknesses:**

**Strengths**
- The overall math an intuition behind the approach is well thought out, and exceedingly interesting and insightful. Unlike many modern distillation methods, this approach is novel and is rooted in both empirical observations and theoretical insights that make for a compelling distillation method.
- The presented analysis in 6.1 is very interesting, and constitutes a thoughtful empirical analysis of the proposed KDMs.

**Weaknesses**
- I think the overall empirical results are a bit weak. The authors make clear emphasis the delineation between online and offline methods, but I think this emphasis is overstated and somewhat attempts to cover their weaker results.
- Especially on unconditional CIFAR-10, KDMs barely beat *any* other distillation method, besides their chosen baseline of GET. The authors also incorrectly portray 2-rf++ as just a continuous flow method, when in fact **it is also a form of offline distillation** if I'm not mistaken.
- Additionally, the relevant offline distillation methods that the authors *should* be comparing against, ADD and LADD, are generally done on much large Text2Image models rather than these small tasks (as GET is not particularly adopted if I'm not mistaken), which makes it hard to assess the overall future performance of KDMs, and whether such a framework could be competitive with these primarily adversarial methods, which are the most common forms of offline distillation.


**Final Decision**: Overall, I do not think this paper is ready for publication yet. While I thoroughly enjoyed the methodological contribution and found these insights exciting and potentially useful for the wider acceleration community, it is hard to ignore the overall poor performance of the approach, and thus in all the paper seems like an interesting, but generally less performant distillation method. I would recommend the authors attempt what they can to either scale up their experiments, continue to explore the empirical design space of the method to improve performance, or recreate existing popular distillation methods at the scale feasible for their compute budget to demonstrate the strengths of KDMs.

---

> ### Author Rebuttal · Authors · 2025-07-30
>
> Thank you for your thoughtful and detailed review. We appreciate your recognition of the novelty and theoretical depth of our approach, as well as your engagement with our analysis in Section 6.1. Your feedback highlights both the strengths of our framework and important areas for further development. We address your points in detail below. Given the opportunity, we will incorporate our modifications into a final revision.
>
> ---
>
> > W1: *I think the overall empirical results are a bit weak. The authors make clear emphasis the delineation between online and offline methods, but I think this emphasis is overstated and somewhat attempts to cover their weaker results.*
>
> We respectfully disagree with the characterization that our emphasis on the offline setting is meant to obscure weaker results. Our goal is to highlight the distinctive general advantages of offline distillation compared to online distillation. Namely, reduced teacher NFEs, lower memory requirements, and improved privacy (as discussed in lines 36–42). While our method does not yet match the performance of leading online techniques, it currently achieves state-of-the-art results within the offline setting, and the gap is steadily narrowing. As shown in Tables 1–3 and our accompanying discussion, we are transparent about where our method stands relative to other paradigms. Part of our motivation is to encourage further attention to offline distillation, a promising yet underexplored direction where our results already demonstrate improved performance.
>
> ---
>
> > W2: *Especially on unconditional CIFAR-10, KDMs barely beat any other distillation method, besides their chosen baseline of GET. The authors also incorrectly portray 2-rf++ as just a continuous flow method, when in fact it is also a form of offline distillation if I'm not mistaken.*
>
> We would like to clarify a potential misunderstanding regarding 2-rf++. While the reviewer suggests it as an offline distillation method, 2-rf++ is an improvement over rectified flow, a simulation-free flow model conceptually closer to flow matching than to knowledge distillation. Unlike distillation methods, it does not learn from a pre-trained teacher model and does not use a distillation loss. Although 2-rf++ generates synthetic data (as we do), it also performs multiple forward passes through the model to fit rectified trajectories (see their Tab. 6), making it fundamentally different from offline distillation approaches, which rely on fixed teacher supervision. For these reasons, we did not include it among the offline distillation baselines.
>
> More generally, there might be a misunderstanding regarding the differences between offline and online settings. In the offline distillation setting, the student model is trained solely on precomputed supervision (e.g., synthetic data) without access to the teacher during optimization. This contrasts with online distillation, where the student is continuously guided by the teacher through live queries or gradient-based feedback during training. Offline distillation offers several inherent advantages in terms of privacy and scalability (please see our discussion with Reviewer `mkU1` for further details regarding these benefits and trade-offs).
>
> ---
>
> > W3: *Additionally, the relevant offline distillation methods that the authors should be comparing against, ADD and LADD, are generally done on much large Text2Image models rather than these small tasks (as GET is not particularly adopted if I'm not mistaken), which makes it hard to assess the overall future performance of KDMs, and whether such a framework could be competitive with these primarily adversarial methods, which are the most common forms of offline distillation.*
>
> We would like to clarify that neither ADD nor LADD qualifies as an offline distillation method under its standard definition. ADD penalizes a distillation loss against the teacher during training (see their Fig. 2), requiring teacher access throughout. LADD, while using synthetic data, also supervises the student using the teacher’s latent features during training. In contrast, our method trains solely on precomputed synthetic samples without teacher interaction, adhering strictly to the offline setting. Therefore, these methods are not directly comparable baselines for our framework.
>
> ---
>
> > Limitations: *The authors do recognize the limitations of their performance, though I do think the overall poor performance is a bit understated and downplayed.*
>
> We appreciate the reviewer’s acknowledgment that we discuss our limitations. However, we would like to re-emphasize that our method achieves state-of-the-art results within the offline distillation setting which differs from the online setting. While it does not yet match the performance of online or simulation-free approaches, we are transparent about this gap throughout the paper and emphasize it to contextualize our contributions. Our intention is not to downplay limitations, but rather to highlight the strengths and practical trade-offs of offline distillation, and to encourage further exploration in this emerging area. We are open to revise our paper to reflect this better, following reviewer suggestions.
>
> ---
>
> > Final Decision 1: *... overall poor performance of the approach, and thus in all the paper seems like an interesting, but generally less performant distillation method.*
>
> We have addressed the performance concerns in detail (see W1 and Limitations), emphasizing that our method is state-of-the-art within the offline setting and closing the gap with other paradigms. We hope these clarifications help contextualize the results and encourage a reconsideration of the overall assessment.
>
> ---
>
> > Final Decision 2: *... I would recommend the authors attempt what they can to either scale up their experiments, continue to explore the empirical design space of the method to improve performance, or recreate existing popular distillation methods at the scale feasible for their compute budget to demonstrate the strengths of KDMs.*
>
> We agree that scaling and further exploration are promising directions and appreciate the constructive suggestions. As mentioned earlier, our current results already show strong promise in the offline setting. We hope our clarifications above will help demonstrate the strengths of KDMs within this context and support a more favorable evaluation.

---

> ### Comment · Reviewer_zaaq · 2025-08-01
>
> I appreciate the concerted response from the authors.
>
> >**More generally, there might be a misunderstanding regarding the differences between offline and online settings. In the offline distillation setting, the student model is trained solely on precomputed supervision (e.g., synthetic data) without access to the teacher during optimization. This contrasts with online distillation, where the student is continuously guided by the teacher through live queries or gradient-based feedback during training.**
>
> I'm not sure if I agree with this, as here in the authors' rebuttal it seems that their definition of what constitutes "offline" distillation is more restrictive than both what I believed to be true and what is stated in the paper. In the paper (lines 98-99), the authors state that "Online distillation techniques [44, 58] supervise the student directly with teacher predictions during training"; this definition makes sense to me, as it hinges upon receiving supervision signal from teacher ***predictions*** during training (i.e. using some transformation of the teacher's standard outputs as a score model to provide training signal). However, the authors argue in the rebuttal that techniques like LADD (I apologize for my incorrect idea about ADD being offline) are not sufficiently offline because of the use of teacher **features** during training, as the discriminator in LADD is built off of the intermediate teacher features (I would not consider these to be predictions). I respectfully disagree with this restrictive definition of "offline" distillation, as many techniques such as LADD, Seaweed-APT, or UFO-Gen would not count as offline distillation purely by the initialization of their discriminator with the teacher backbone, despite the fact that the teacher never applies a direct distillation loss in these settings. In fact, one could easily extend this restrictive definition of offline distillation to exclude *anything* besides learning from generated (noise, sample) pairs, which would even invalidate the use of the discriminator in the present work. I understand, however, that this is mostly arguing semantics, and that the authors reported definition in the rebuttal aligns more with their goals of privacy.
>
> More importantly, I think there is some misunderstanding regarding the comparison to 2-rf++.
>
> >**We would like to clarify a potential misunderstanding regarding 2-rf++. While the reviewer suggests it as an offline distillation method, 2-rf++ is an improvement over rectified flow, a simulation-free flow model conceptually closer to flow matching than to knowledge distillation... Although 2-rf++ generates synthetic data (as we do), it also performs multiple forward passes through the model to fit rectified trajectories (see their Tab. 6), making it fundamentally different from offline distillation approaches, which rely on fixed teacher supervision**
>
>  Upon further investigation of the 2-rf++ paper, I am relatively confident that it *is* an offline distillation method. The authors claim that "While the reviewer suggests it as an offline distillation method, 2-rf++ is an improvement over rectified flow, a simulation-free flow model conceptually closer to flow matching than to knowledge distillation" does not make much logical sense: comparing flow matching to knowledge distillation is a bit of an apples to oranges comparison, rectified flows (or more specifically, the ReFlow process) is a *form* of knowledge distillation *applied* to flow matching (which is equivalent to diffusion under specific parameterizations). Most saliently, the claim that "[2-rf++]  also performs multiple forward passes through the model to fit rectified trajectories (see their Tab. 6)" does not seem to be true. Their Tab. 6 denotes the **total** number of forward passes from both the teacher and student during offline data generation (where only the teacher is used) and distillation (where only the student is used). Even if that was true, it is unclear why "multiple forward passes through the model to fit rectified trajectories" would make it any less offline by the authors' own definition. Thus, I am reasonably confident 2-rf++ is an offline method, and thus, KDMs could not be considered SOTA for offline methods. I will hence maintain my score.

---

> > ### Author Response · Authors · 2025-08-05
> > **Discussion Cont. - 3**
> >
> > ### Reviewer Comment 2 - Cont.:
> >
> > Following the above discussion, and to broaden the context of our work, we have completed the evaluation of **rf-2++ without a pre-trained model**, following a more comparable setting. We also collected the corresponding **Inception Score (IS)**. The results are summarized below:
> >
> > | Method                           | FID Score ↓ | IS Score ↑ |
> > | -------------------------------- | ----------- | ---------- |
> > | rf-2++ – with pre-trained EDM    | 3.07        | 9.12       |
> > | rf-2++ – without pre-trained EDM | 6.32        | 9.01       |
> > | GET                              | 6.91        | 9.16       |
> > | KDM                              | 4.97        | 9.22       |
> >
> > Running **rf-2++** without a pre-trained EDM results in a degradation of approximately **3 FID points** and **0.11 in IS score**, positioning its performance between **KDM** and **GET**. Notably, **KDM** achieves a higher IS score than both versions of **rf-2++** evaluated in this comparison. We’ll incorporate these results in the final version and sincerely thank you for the constructive feedback. We hope the findings will be helpful for a reassessment and are happy to address any further questions.

---

> ### Author Response · Authors · 2025-08-03
> **Discussion Cont. - 1**
>
> We sincerely appreciate the engagement and thoughtful discussion, which helps drive the work forward. After a careful review of the materials, we would like to continue the discussion.
>
> ### Reviewer Comment 1:
>
> > *It seems that their definition of what constitutes "offline" distillation is more restrictive than both what I believed to be true and what is stated in the paper.*
>
> Thank you for the opportunity to clarify this point. We would like to emphasize that we adhered to the standard protocol suggested by [1] to ensure a fair comparison between our method and competing offline approaches. We would like to emphasize that our work already includes the following definition for offline distillation, consistent with [1]:
> *Offline distillation relies on precomputed noise-image pairs generated by the teacher, and the student learns without further querying the teacher model.*
>
>
> ---
>
> > *In fact, one could easily extend this restrictive definition of offline distillation to exclude anything besides learning from generated (noise, sample) pairs, which would even invalidate the use of the discriminator in the present work.*
>
> We believe this comment may reflect a misunderstanding of our method. Crucially, our discriminator is trained solely on the final samples from the precomputed (noise, sample) pairs and does not use teacher predictions, intermediate features, or adversarial signals. This distinguishes our approach from methods like 2-rf++, LADD, Seaweed-APT, or UFO-Gen, which rely on teacher-internal features or outputs during training. Including such supervision would compromise the fairness of comparisons with other purely offline methods such as GET.
>
> While our approach may appear restrictive in that it prohibits using pretrained teacher models, we believe this constraint enhances the method's adaptability. Relying on pretrained weights often couples the student to the teacher’s architecture and scale, whereas our approach enables flexible, resource-efficient deployment across a wider range of scenarios.
>
> ---
>
> > *I understand, however, that this is mostly arguing semantics, and that the authors’ reported definition in the rebuttal aligns more with their goals of privacy.*
>
> We appreciate the constructive discussion and are glad to have reached a shared understanding. We also thank the reviewer for prompting this exchange, which allowed us to highlight additional strengths of our offline setting beyond privacy. While privacy is certainly a key benefit of avoiding teacher supervision, it is not the only one. More broadly, removing dependence on the teacher model enables greater flexibility in the student’s architectural design and model size.
>
> Unlike methods such as 2-rf++, which must re-use or adapt the teacher model's architecture and parameter scale (see answer for Comment 2), approaches based on (noise, sample) pairs are decoupled from the teacher. This allows, for example, distilling a 7B-parameter Transformer-based teacher into a much smaller and differently structured student. In contrast, 2-rf++ relies on initializing with a pretrained model and, as a result, is tied to the teacher’s architecture and capacity. KDM and similar approaches enable smaller, more versatile student models, which we believe is important for practical and scalable deployment.

---

> ### Author Response · Authors · 2025-08-03
> **Discussion Cont. - 2**
>
> ### Reviewer Comment 2:
>
> We respectfully disagree with the characterization of 2-rf++ as an offline distillation method in the discussed context.
>
> As stated by the authors at the beginning of Section 5 of the *2-rf++* paper:
>
> > *"We call these combined improvements to Reflow training 2-rectified flow++ ..."*
>
> These improvements refer to the four components outlined in Section 4, which details their method and includes:
>
> * **4.1** Timestep distribution,
> * **4.2** Loss function,
> * **4.3** **Initialization with pre-trained diffusion models**, and
> * **4.4** Incorporating real data.
>
> Focusing on Section 4.3, the authors write:
>
> > *"Proposition 1 allows us to initialize the Reflow with the pre-trained diffusion models such as EDM \[Karras et al., 2022] or DDPM \[Ho et al., 2020]..."*
>
> and later:
>
> > *"Starting from the vanilla 2-rectified flow setup \[Liu et al., 2022] (config A), we initialize 1-rectified flow with the pre-trained EDM (VE)..."*
>
> This demonstrates that 2-rf++ directly accesses and leverages the teacher model’s weights during training, which violates the previously discussed definition. In contrast, our method (KDM) strictly adheres to the discussed offline setting, it never uses the teacher during student training and relies only on precomputed data. Therefore, we respectfully disagree that 2-rf++ qualifies as an offline method under our setting. Consequently, we maintain that KDM can be fairly considered state-of-the-art among **offline** approaches.
>
> Furthermore, we would like to clarify our statement regarding the forward passes and apologize for any misunderstanding. While Table 6 in the 2-rf++ paper lists the number of forward passes for data generation and student training, in our response, our intention was to highlight that this setup is not equivalent to querying a frozen dataset. Rather, the student is initialized with a pretrained teacher model, meaning teacher knowledge is embedded in the student from the outset, which contradicts the offline setting.
>
> Additionally, the reported performance gains of 2-rf++ are significantly influenced by this initialization. The authors report a nearly 5-point improvement in FID (from 12.21 to 7.14) resulting from starting with a pretrained model. However, they do not provide a clean ablation study to isolate the specific contribution of this factor, as results are presented cumulatively. This makes direct comparison with fully offline methods such as KDM and GET, both of which do not rely on teacher initialization, less straightforward. We also note that in Table 3 of the 2-rf++ paper, Inception Scores (IS) are omitted. While the 3-rectified flow (online) variant achieves a better FID (3.96 with 104 steps), it reports an IS of only 9.01, worse than both our method (9.22) and GET (9.16). These omissions and discrepancies further complicate a fair comparison. Nevertheless, we do like to show and give a broader context, therefore, we are working on make a comparison and we are running 2-rf++ without loading the pre-trained model and hope to report results soon.
>
> Finally, as noted earlier, reusing the teacher model constrains methods like 2-rf++ to the teacher's architecture and parameter scale. This limits the flexibility to reduce model size or adapt the architecture for different use cases. In contrast, KDM and GET are fully decoupled from the teacher model, enabling greater adaptability and efficiency, especially in resource-constrained environments.
>
> We hope this clarification is helpful and that it may lead you to reconsider your assessment. We would be happy to incorporate this explanation into the final version of the paper to improve clarity and transparency, and we welcome any further questions or feedback.
>
>
> **\[1]** One-Step Diffusion Distillation via Deep Equilibrium Models — Zhengyang Geng, Ashwini Pokle, J. Zico Kolter.

---

> ### Author Response · Authors · 2025-08-07
> **Final Notes**
>
> We would like to thank you once again for engaging in a discussion that has helped move our work forward. As the rebuttal comes to a close, we’re happy to clarify any remaining questions you might have.
>
> Additionally, in light of the ongoing review, we’d like to draw your attention to new results shared in the latest comment to reviewer FqCX. The reviewer suggested a comparable setup and method for our offline setup. In response, we conducted an additional experiment comparing KDM with a state-of-the-art one-step generation method, suggested by the reviwer, using the same generated data and training budget.
>
> We trained an IMM [1] model from scratch using only the data (excluding the noise) from the data-noise pairs, and additionally computed the Inception Score (IS), which was not originally reported. Methods were trained under the same compute budget of 102.4 million images. The results are presented below:
>
> | Method                    | FID ↓ | IS ↑ |
> | ------------------------- | ----- | ---- |
> | IMM (with generated data) | 4.81  | 9.17 |
> | GET                       | 6.91  | 9.16 |
> | KDM                       | 4.97  | 9.22 |
> | KDM-F                     | 4.68  | 9.08 |
>
> These results demonstrate that, under the same generated data and budget settings, our method (KDM) is competitive with state-of-the-art approaches for one-step generation from scratch. This supports our claim that KDM is a promising and viable alternative for this task. Finally, following our in-depth discussions, we will revise the statement in the paper from **state-of-the-art** to **competitive**, as this more accurately reflects the current context after the rebuttal.
>
> We will include these new results in the final revision alongside the others. We sincerely thank you once again for your constructive feedback, which has greatly helped us improve our work.
>
> ---
>
> [1] Inductive Moment Matching

---

### Official Review · Reviewer_dXrx · 2025-07-01

**Clarity:** 3
**Significance:** 3
**Originality:** 3
**Rating:** 5
**Confidence:** 3

**Summary:**

This paper introduces the Koopman Distillation Model (KDM), a novel method for offline distillation of diffusion models. The main innovation is its use of Koopman operator theory to represent the non-linear process of generating an image from noise in diffusion models as a simple linear transformation in a learned embedding space. This provides  a principled way to achieve single-step generation.

The authors provide theoretical proofs showing that the diffusion process can be accurately represented by a finite-dimensional Koopman operator and that this representation preserves the semantic structure of the generated images. Empirically, KDM outperforms previous offline distillation methods on standard benchmarks like CIFAR-10, FFHQ, and AFHQv2, improving FID scores by up to 40% in a single generation step. The method is not only fast, achieving a 4x speedup in training time compared to prior work, but also scales effectively with model size.

**Questions:**

I would appreciate it if the authors can answer the below questions from the weaknesses section above:

- Point 2. from "Contribution of the $L_{\text{Koopman}}$ loss".
- Point 1. from "Weighting of the $L_{\text{Koopman}}, L_{\text{adv}}$ losses".
- Point 1. from "Practical relevance of the bound in Theorem 5.2".

**Ethical Concerns:**

["NO or VERY MINOR ethics concerns only"]

**Final Justification:**

The authors have adequately answered all of my questions and I have therefore decided to retain my score. Upon reading the comments of other reviewers (in particular comments about significance of offline distillation and empirical results) I have decided to revise my confidence.

**Limitations:**

Yes.

**Paper Formatting Concerns:**

NA.

**Quality:**

3

**Strengths And Weaknesses:**

# Strengths:

### **Quality**:
Overall, the paper is of high quality and supports its claims with extensive empirical validation, which includes experiments on standard benchmark datasets (CIFAR-10, FFHQ, AFHQv2) using standard evaluation metrics (FID, IS) as well as a human evaluation study.

### **Clarity**:
The paper is well-written and clear. The paper is logically structured and the method is described in detail, including the specific loss function and model architecture.

### **Significance**:
By developing a method for single-step offline distillation that is not only fast but also achieves state-of-the-art results, the paper provides
provides a practical tool that both researchers and practitioners are likely to use and build upon. The efficiency gains are significant, with a reported 4x speedup in training and over 8x faster sampling compared to the previous offline SOTA.

### **Originality**:
This paper presents a original idea: it uses Koopman operator theory to linearize the complex generation dynamics of diffusion models, leading to a novel approach for offline distillation. Moreover, the insight that the semantic structure in the noise-to-image mapping of diffusion models is preserved in the learned embedding of the Koopman operator is an original and valuable contribution in its own right.

---

# Weaknesses:

### **Contribution of the  $L_{\text{Koopman}}$ loss**:

- It's not clear to what extent the $L_{\text{Koopman}}$ loss contributes to the success of the approach. The authors include an adversarial loss "to further refine the distillation process", however without the adversarial component (see Table 5) their method actually performs worse than their main point of comparison (GET).
- Moreover, GET itself is not adversarially trained so the comparison between an adversarially trained KDM and a non-adversarially trained GET is not totally fair. The authors mention in Appendix D.3 that they struggled to train GET with an adversarial loss - it would be helpful to also see the results/loss curves for these attempts.

### **Weighting of the $L_{\text{Koopman}}, L_{\text{adv}}$ losses**:

- Following up from the previous comments, I wonder if the authors tried different weightings for the $L_{\text{Koopman}}$ and  $L_{\text{adv}}$ losses? Currently, the weighting on the different components of $L_{\text{Koopman}}$ is uniform, even though they may potentially lie on different scales during training. Moreover, there is no ablation for different values of $\lambda_{\text{adv}}$. It would be nice to see some ablations for this to better understand the effect of the adversarial loss and also the different components of $L_{\text{Koopman}}$.

### **Practical relevance of the bound in Theorem 5.2**:

- Thm. 5.2 establishes that semantic proximity is preserved, showing that the distance between two generated images is bounded by the distance between their latent representations. However, clearly the quality of this bound depends on the norm $||C_\eta||$ - can the authors show what kind of norms $||C_\eta||$ actually achieves in practise?

---

> ### Author Rebuttal · Authors · 2025-07-30
>
> Thank you for your thoughtful and constructive review. We appreciate your recognition of the novelty, clarity, and potential impact of our work, as well as your detailed feedback on areas for improvement. We address your questions below and, given the opportunity, will incorporate your suggestions into the final revision.
>
> ---
>
> > Contribution of the $L_{\text{Koopman}}$ loss: *Moreover, GET itself is not adversarially trained so the comparison between an adversarially trained KDM and a non-adversarially trained GET is not totally fair. The authors mention in Appendix D.3 that they struggled to train GET with an adversarial loss - it would be helpful to also see the results/loss curves for these attempts.*
>
> Following our original submission, we further explored how to incorporate adversarial training into GET and were able to successfully adapt it. On CIFAR-10, the adversarially trained GET achieves an FID of **6.13**, an improvement over the original GET’s 6.91, though still underperforming compared to our KDM’s **4.68**. Regarding the training dynamics, we observed a stable behavior in the loss curves during adversarial training:
>
> | Step           | 0%     | 10%    | 20%    | 30%    | 40%    | 50%    | 60%    | 70%    | 80%    | 90%    | 100%   |
> | :------------- | :----- | :----- | :----- | :----- | :----- | :----- | :----- | :----- | :----- | :----- | :----- |
> | KDM Gen. Loss  | 1.98   | 0.74   | 0.65   | 0.71   | 0.75   | 0.84   | 0.68   | 0.79   | 0.68   | 0.89   | 0.84   |
> | KDM Disc. Loss | 1.00   | 1.15   | 1.12   | 1.08   | 1.17   | 1.05   | 1.02   | 1.14   | 1.15   | 1.09   | 1.17   |
> | GET Gen. Loss  | 1.95   | 0.89   | 0.75   | 0.77   | 0.72   | 0.72   | 0.74   | 0.88   | 0.85   | 0.94   | 0.91   |
> | GET Disc. Loss | 1.01   | 1.09   | 0.99   | 1.05   | 1.15   | 1.06   | 1.17   | 1.12   | 1.07   | 1.13   | 1.19   |
> | CTM Gen. Loss  | -      | -      | -      | -      | -      | -      | -      | -      | -      | -      | -      |
> | CTM Disc. Loss | 0.025  | 0.009  | 0.008  | 0.009  | 0.010  | 0.008  | 0.008  | 0.011  | 0.010  | 0.012  | 0.012  |
>
> *Table: Convergence of training loss over uniformly sampled steps.*
>
> These results suggest that while adversarial training does enhance GET’s generative quality, KDM provides additional benefits beyond what adversarial objectives alone can achieve.
>
> Note, the CTM framework computes the adversarial loss differently and on a different scale, as it employs a distinct discriminator architecture and adversarial setup. CTM released their adversarial loss values, which we also report in the table below for completeness. While the fidelity is less accurate compared to our method and GET since the loss is combined for both the generator and the discriminator, the reported loss exhibits similarly stable behavior, as expected, showcasing the robustness of their model.
>
> ---
>
> > Weighting of the $L_{\text{Koopman}}, L_{\text{adv}}$ losses: *Following up from the previous comments, I wonder if the authors tried different weightings for the $L_{\text{Koopman}}$ and $L_{\text{adv}}$ losses? Currently, the weighting on the different components of $L_{\text{Koopman}}$ is uniform, even though they may potentially lie on different scales during training. Moreover, there is no ablation for different values of $\lambda_{\text{adv}}$. It would be nice to see some ablations for this to better understand the effect of the adversarial loss and also the different components of $L_{\text{Koopman}}$.*
>
> We explore different weightings of the Koopman components $L_{\text{rec}}, L_{\text{lat}}$ and $L_{\text{pred}}$. The results indicate that the model is relatively robust to variations in these weights, with FID scores remaining in a narrow range (4.94–5.03).
>
> In addition, we evaluated the effect of the adversarial loss weight $\lambda_{\text{adv}}$, where we found that $\lambda_{\text{adv}} = 1.0$ yielded the best FID (4.57), compared to 4.98 for $\lambda_{\text{adv}} = 0.5$ and 4.67 for the default 0.25 setting reported in the main paper. We will include these results and clarify them further in the appendix to improve transparency.
>
> ---
>
> >Practical relevance of the bound in Theorem 5.2: *Thm. 5.2 establishes that semantic proximity is preserved, showing that the distance between two generated images is bounded by the distance between their latent representations. However, clearly the quality of this bound depends on the norm $\| C_\eta \|$ - can the authors show what kind of norms $\| C_\eta \|$ actually achieves in practise?*
>
> In practice, we observe an average Koopman operator spectral norm of 0.88 with std 0.18, which results in a reasonably tight bound and supports the practical utility of the theorem. Furthermore, our framework explicitly allows for controlling this norm by penalizing the eigenvalues of the Koopman operator. As discussed in Appendix C, the Koopman operator is factorized in a way that enables direct constraint of its spectrum. This provides a principled means to regularize $\| C_\eta \|$, thereby improving the semantic consistency of generated samples while maintaining stability and interpretability.

---

> > ### Comment · Reviewer_dXrx · 2025-08-02
> > **Response to authors**
> >
> > Thank you for addressing my questions.
> >
> > I do not believe that the reported numbers for GET display stable behaviour - simply plotting these shows different. However, I believe that the authors have made enough effort to incorporate adversarial training into GET for this not to be a major concern.
> >
> > I have decided to retain my score. After having read comments from other reviewers, I have also decided to revise my confidence rating.

---

> > > ### Author Response · Authors · 2025-08-05
> > > **Thank you**
> > >
> > > Thank you for engaging with us and helping to improve our work.

---

### Official Review · Reviewer_mkU1 · 2025-07-03

**Clarity:** 3
**Significance:** 3
**Originality:** 3
**Rating:** 5
**Confidence:** 4

**Summary:**

This work proposes the Koopman distillation model for offline distillation, in order to compress diffusion models. The authors observed that there is lack of theoretical adoption of dynamic systems theory to diffusion models. It is also empirically found that there is semantic coherence in latent spaces induced by diffusion models. From these two observations, Koopman distillation model (KDM) is proposed and theoretically analyzed. It distills diffusion models via linear evolution in a learned embedding space which is generated by an encoder given a noisy input. From the embedding, a decoder then reconstructs the clean samples. The authors also theoretically proved the existence of finite-dimensional Koopman representations and semantic structure preservation under mild assumptions. Experimental results also demonstrate its performance compared with baselines.

**Questions:**

1. *Training loss*: The design of the loss function contains the adversarial loss in order to enhance the quality of the generated images. However, it may introduce the instability of the training process and the difficulty to converge. Would the authors demonstrate the training loss convergence curve in their experiments and compare it with GET and CTM?

2. *Memory cost*: In experiments, the quality of generated images is compared on the same scale of parameters across KDM and GET. However, as claimed in the introduction that KDM as an offline distillation method is more efficient than diffusion distillation methods such as CTM. Hence could the comparison include CTM on the parameter efficiency or other metrics to demonstrate the efficiency of KDM over CTM?

3. *Higher resolution dataset*: The experiments are conducted on CIFAR-10, FFHQ 64×64 and AFHQv2 64×64 datasets. Would the authors present the experiments on ImageNet-1k to evaluate the performance of KDM on higher resolution datasets?

4. *Clarification*: A minor question is on the notations. Could the authors clarify and explain further on the differences between $C_\mu$ and $C_\eta$?

5. *Quality*: The quality and motivations should be stronger. The current prominent compression method for diffusion models are consistency models and their variants. Although the privacy is mentioned as one of the advantages, experiments do not demonstrate and support it. It would improve the quality to include them and articulate them in experiments briefly.

**Ethical Concerns:**

["NO or VERY MINOR ethics concerns only"]

**Final Justification:**

All my questions are well addressed by the authors. Hence I raised my score from 4 Borderline Accept to 5 Accept.

**Limitations:**

Yes.

**Quality:**

2

**Strengths And Weaknesses:**

**Strengths**
1. *Originality*: This work novelly proposed the one-step Koopman distillation model (KDM) as an offline distillation of diffusion models. It is of originality to consider the Koopman operator in distillation of diffusion models, which did not appear in past research.

2. *Clarity*: The presentation is clear and logically linear. The whole work is easy to follow. The motivations and methods are clearly articulated in terms of the current limitations of diffusion models and the mechanism and pipeline of KDM.

3. *Significance*: The excessive computational cost is a current pressing issue of diffusion models as a powerful visual content generator. The proposed KDM is important to solve this problem which is also vital.

**Weaknesses**

1. *Quality*: The performance of the proposed KDM does not of the state-of-the-art among all the presented compression methods for diffusion models. Although KDM is an offline distillation method, the experimental results did not demonstrate the advantages of it over CTM as diffusion distillation methods.

---

> ### Author Rebuttal · Authors · 2025-07-30
>
> Thank you for your thoughtful and constructive review. We appreciate your recognition of the novelty, clarity, and significance of our proposed KDM, as well as your detailed feedback. Your comments on evaluation, training stability, efficiency comparisons, and clarity are insightful. We address your questions below and, if given the opportunity, will incorporate your suggestions into the final revision.
>
> ---
>
> > W1: Quality: The performance of the proposed KDM does not of the state-of-the-art among all the presented compression methods for diffusion models. Although KDM is an offline distillation method, the experimental results did not demonstrate the advantages of it over CTM as diffusion distillation methods.
>
> While KDM does not yet match the performance of online or simulation-free methods like CTM, we are transparent about this gap throughout the paper and use it to contextualize our contributions. Our aim is not to claim superiority over all alternatives, but to establish a strong baseline within the offline distillation setting and demonstrate its practical viability.
>
> Specifically, although CTM reports strong results, it is an online distillation method that requires repeated teacher queries during training, resulting in approximately 6.5x higher computational cost than KDM (5.2M vs. 0.8M NFEs using batches of 128 samples on CIFAR-10). In contrast, KDM operates entirely in the offline setting using precomputed supervision, making it more scalable and resource-efficient. Additionally, KDM is significantly lighter in terms of model size, using only $\approx65$M parameters (our Koopman and GAN models), while CTM requires $\approx180$M parameters, comprising both a full EDM teacher and student, as well as an auxiliary state-of-the-art GAN ($\approx30$M parameters). Furthermore, while KDM employs a simple GAN framework, CTM uses a state-of-the-art GAN that is more than 15x larger in size compared to ours. Though both methods benefit from adversarial losses (as shown in CTM’s Table 3), they are not direct competitors due to these fundamental differences in training setups, supervision, compute, and model size.
>
> Finally, we would like to emphasize the privacy advantage of KDM over CTM as an offline distillation method. Offline distillation is limiting vulnerability to inversion or membership inference attacks. In traditional online distillation, especially when teacher models are repeatedly queried (e.g., denoising at various timesteps), there's a risk that an adversary could recover training data or determine whether a sample was in the original dataset.
>
> In contrast, offline distillation relies only on a fixed dataset of noise–data pairs. Once these are generated (and potentially anonymized or filtered), the teacher model can be discarded or kept private. This one-time generation significantly reduces the attack surface, as no further queries are made to the teacher, and thus it becomes computationally infeasible for the student model, or an adversary accessing it, to reconstruct the original training distribution beyond what the static examples reveal.
>
> As a simple example of a use case, it's possible to outsource the training process to a third party without exposing sensitive model parameters or proprietary intellectual property. The curator (e.g., the data owner) can pre-select and sanitize the training samples to remove personally identifiable information (PII) or domain-sensitive content. Since the third-party trainer doesn’t need access to the full generative model or the underlying sampling process, they are not exposed to unintended internal correlations or behaviors learned by the teacher model, which may inadvertently reveal private training data. As a result, sensitive institutions (like hospitals, finance firms, or proprietary content holders) can leverage external compute without compromising confidentiality.
>
> ---
>
> > Q1: Training loss: The design of the loss function contains the adversarial loss in order to enhance the quality of the generated images. However, it may introduce the instability of the training process and the difficulty to converge. Would the authors demonstrate the training loss convergence curve in their experiments and compare it with GET and CTM?
>
> To assess the training stability of KDM, we provide a comparison of the generator and discriminator loss curves for both KDM and GET across uniformly sampled training steps (see the table below). Despite the inclusion of an adversarial loss in KDM, the generator and discriminator losses exhibit stable behavior and convergence, suggesting that the adversarial component does not introduce significant instability in practice.
>
> The CTM framework computes the adversarial loss differently and on a different scale, as it employs a distinct discriminator architecture and adversarial setup. CTM released their adversarial loss values, which we also report in the table below for completeness. While the fidelity is less accurate compared to our method and GET since the loss is combined for both the generator and the discriminator, the reported loss exhibits similarly stable behavior, as expected, showcasing the robustness of their model.
>
> | Step           | 0%     | 10%    | 20%    | 30%    | 40%    | 50%    | 60%    | 70%    | 80%    | 90%    | 100%   |
> | :------------- | :----- | :----- | :----- | :----- | :----- | :----- | :----- | :----- | :----- | :----- | :----- |
> | KDM Gen. Loss  | 1.98   | 0.74   | 0.65   | 0.71   | 0.75   | 0.84   | 0.68   | 0.79   | 0.68   | 0.89   | 0.84   |
> | KDM Disc. Loss | 1.00   | 1.15   | 1.12   | 1.08   | 1.17   | 1.05   | 1.02   | 1.14   | 1.15   | 1.09   | 1.17   |
> | GET Gen. Loss  | 1.95   | 0.89   | 0.75   | 0.77   | 0.72   | 0.69   | 0.74   | 0.88   | 0.85   | 0.94   | 0.91   |
> | GET Disc. Loss | 1.01   | 1.09   | 0.99   | 1.05   | 1.15   | 1.06   | 1.17   | 1.12   | 1.07   | 1.13   | 1.19   |
> | CTM Gen. Loss  | -      | -      | -      | -      | -      | -      | -      | -      | -      | -      | -      |
> | CTM Disc. Loss | .025   | .009   | .008   | .009   | .010   | .008   | .008   | .011   | .010   | .012   | .012   |
>
> *Table: Convergence of training loss over uniformly sampled steps.*
>
> ---
>
> > Q2: Memory cost: In experiments, the quality of generated images is compared on the same scale of parameters across KDM and GET. However, as claimed in the introduction that KDM as an offline distillation method is more efficient than diffusion distillation methods such as CTM. Hence could the comparison include CTM on the parameter efficiency or other metrics to demonstrate the efficiency of KDM over CTM?
>
> We have addressed this comparison in our response above, where we clarify that KDM uses significantly fewer parameters ($\approx65$M, corresponding to our Koopman+GAN models), whereas CTM requires $\approx180$M parameters due to the inclusion of both a teacher and student EDM, as well as an auxiliary GAN ($\approx30$M). These distinctions underline the efficiency advantages of the offline KDM framework in both memory aspects.
>
> ---
>
> > Q3: Higher resolution dataset: The experiments are conducted on CIFAR-10, FFHQ 64x64 and AFHQv2 64x64 datasets. Would the authors present the experiments on ImageNet-1k to evaluate the performance of KDM on higher resolution datasets?
>
> Due to the significant computational resources required for training on ImageNet-1k at scale, we were unfortunately unable to include these experiments in the current work. Our focus was instead on widely used, moderately sized benchmarks such as CIFAR-10, FFHQ 64x64, and AFHQv2 64x64, which still offer meaningful diversity in image content and structure. We view evaluation on higher-resolution datasets as an important direction for future work, particularly as greater resources become available.
>
> ---
>
> > Q4: Clarification: A minor question is on the notations. Could the authors clarify and explain further on the differences between $C_\mu$ and $C_\eta$?
>
> In our framework, $C_\eta$ denotes the primary linear (Koopman) operator that governs the evolution of latent inputs through the dynamical system. On the other hand, $C_\mu$ is a distinct linear operator that is learned to act solely on the auxiliary condition or context variable, not on the inputs themselves. This structure reflects a modular design often seen in linear control systems, where different operators handle different aspects of the system dynamics and exogenous inputs.
>
> ---
>
> > Q5: Quality: The quality and motivations should be stronger. The current prominent compression method for diffusion models are consistency models and their variants. Although the privacy is mentioned as one of the advantages, experiments do not demonstrate and support it. It would improve the quality to include them and articulate them in experiments briefly.
>
> Our goal is to highlight the unique advantages of offline distillation, particularly its flexibility, efficiency, and broader reuse of precomputed supervision signals, as a complementary direction to online approaches. As noted in Lines 36–42 of the introduction, our discussion of privacy and other benefits refers to general advantages of offline distillation as a paradigm, supported by prior work (e.g., \[Fernandez et al., 2023; Geng et al., 2023]) rather than specific claims about our method. We agree that incorporating explicit empirical evaluations of privacy would strengthen the contribution, but this was outside the scope of our current study. We have revised the text to clarify that these benefits are general to the offline setting and not directly assessed in our experiments.

---

> > ### Comment · Reviewer_mkU1 · 2025-08-05
> >
> > I thank the authors for their efforts and detailed reply. Please see my comments as follows.
> >
> > > Q1 Training loss: The design of the loss function contains the adversarial loss in order to enhance the quality of the generated images. However, it may introduce the instability of the training process and the difficulty to converge. Would the authors demonstrate the training loss convergence curve in their experiments and compare it with GET and CTM?
> >
> > Thank you for your reply and reporting the training loss of the discriminator for KDM, GET and CTM. I plotted them and it seems that except CTM, the training losses do not converge and are not stable. It might be interesting to study the reason behind.
> >
> > > Q3 Higher resolution dataset: The experiments are conducted on CIFAR-10, FFHQ 64x64 and AFHQv2 64x64 datasets. Would the authors present the experiments on ImageNet-1k to evaluate the performance of KDM on higher resolution datasets?
> >
> > Thank you for your explanation. Could you explicitly demonstrate your computing device settings, such as the number of GPUs used and GPU models?
> >
> > For other replies to my previous questions, I have no further comments.

---

> > > ### Author Response · Authors · 2025-08-05
> > > **Response**
> > >
> > > Thank you very much for engaging in the discussion, it has been valuable in helping us improve our work. Regarding your comments:
> > >
> > > >Thank you for your reply and reporting the training loss of the discriminator for KDM, GET and CTM. I plotted them and it seems that except CTM, the training losses do not converge and are not stable. It might be interesting to study the reason behind.
> > >
> > >
> > > We apologize for any confusion. The perceived instability is largely an artifact of the table's sparse sampling and the inherent variance in the loss values. On our platform, the graphs suggest healthy generator–discriminator dynamics: neither model collapses, and both exhibit balanced, competitive behavior. This interplay, where improvements in one model challenge the other, indicates that the GAN is training effectivel. Moreover, these results were consistent across all of our experiments. Unfortunately, we were unable to convey this effectively through the table. Therefore, to improve readability in this rebuttal, we smoothed the loss curve and applied denser sampling at the early stages of training, where it helps reveal the trend more clearly.
> > >
> > > | Step           | 0%  | 0.5% | 1%   | 1.5% | 2%   | 4%   | 10%  | 20%  | 30%  | 40%  | 50%  | 60%  | 70%  | 80%  | 90%  | 100% |
> > > |:---------------|:----|:-----|:-----|:-----|:-----|:-----|:-----|:-----|:-----|:-----|:-----|:-----|:-----|:-----|:-----|:-----|
> > > | KDM Gen. Loss  | -   | 3.91 | 4.05 | 3.84 | 2.90 |  1.14| 0.84 | 0.65 | 0.75 | 0.72 | 0.82 | 0.88 | 0.81 | 0.78 | 0.69 | 0.65 |
> > > | KDM Disc. Loss | -   |  0.24|  0.31|  0.65|  0.78|  0.96| 1.15 | 1.27 | 1.23 | 1.14 | 1.25 | 1.19 | 1.14 | 1.25 | 1.21 | 1.23 |
> > > | GET Gen. Loss  | -   |  4.12| 4.35 |  3.21|  2.14|  1.84| 0.89 | 0.75 | 0.77 | 0.72 | 0.69 | 0.74 | 0.88 | 0.85 | 0.74 | 0.91 |
> > > | GET Disc. Loss | -   |  0.53| 0.69 | 0.71 |  0.94| 1.01 | 1.11 | 1.15 | 1.09 | 1.21 | 1.16 | 1.12 | 1.15 | 1.09 | 1.14 | 1.10 |
> > >
> > > We discarded the 0% value as it contained a corrupted artifact. The updated table now illustrates the trend and pattern we referred to in our previous comments. Thank you for pointing this out and allowing us to clarify the matter.
> > >
> > > > Thank you for your explanation. Could you explicitly demonstrate your computing device settings, such as the number of GPUs used and GPU models?
> > >
> > > We would be happy to share these details. Our experiments were all run on a **single NVIDIA A6000 GPU or RTX4090**. Because of deployment constraints imposed by the institution IT team, we cannot parallelize jobs across multiple A6000s or RTX4090. Consequently, each experiment reported in the paper ran on a single GPU. Nonetheless, our codebase and model architecture fully support distributed training. We will include these technical specifics in the final revision to enhance the paper’s transparency.
> > >
> > > Thank you again for your feedback. We're happy to provide any additional information or clarification if needed.

---

> > > > ### Comment · Reviewer_mkU1 · 2025-08-08
> > > >
> > > > I thank the authors for their response and updating the experimental results.
> > > >
> > > > For the updated generative losses, I can observe that KDM generative loss and GET generative loss both converged to values lower than the initial losses. However, for the discriminative losses, KDM and GET demonstrated that they converged to higher values than the 0.5% values. This seems to be interesting and could the authors explain the possible reasons?

---

> ### Author Response · Authors · 2025-08-08
> **Response**
>
> We would be happy to address this question. The initially low discriminator losses at 0.5% (0.24 for KDM and 0.53 for GET) indicate that **the discriminator** was effective at distinguishing real from generated data in the early stages of training. This is expected, as **the generator** at this point was still untrained, producing outputs that were easy to identify as fake.
>
> As training progressed, the generator improved significantly, as reflected in its decreasing loss values (from over 4.0 to around 0.7–0.9), meaning it became increasingly capable of producing realistic-looking samples that could partially fool the discriminator.
>
> With the generator’s outputs becoming more convincing, the discriminator’s task naturally became harder, leading to an increase in its loss, which converged to around 1.1–1.2. This higher loss indicates that the discriminator could no longer easily separate real from fake data - a hallmark of successful GAN training. Finally, both losses are relatively stabilized, showing that the generator and discriminator were continuously challenging each other. Such a balanced equilibrium helps prevent collapse, drives the generator generation quality forward, and reflects healthy adversarial dynamics.
>
> Thank you again, and we are happy to address any additional questions or clarifications you may have.

---

### Official Review · Reviewer_FqCX · 2025-07-03

**Clarity:** 3
**Significance:** 1
**Originality:** 3
**Rating:** 4
**Confidence:** 3

**Summary:**

This paper proposes the Koopman Distillation Model (KDM), a novel offline distillation framework designed for one-step generation. The proposed method is grounded in Koopman operator theory, a classical mathematical framework that seeks to represent complex nonlinear dynamics as a globally linear operator acting on a transformed space of "observable" functions. In practice, KDM employs an encoder-linear-decoder architecture. A neural network encoder maps a noisy input into a high-dimensional latent space; a learned, finite-dimensional Koopman matrix then propagates the latent state forward in a single, linear step; finally, a decoder network reconstructs the clean, final sample from the evolved latent state. The authors provide theoretical justification for this architecture, proving that the learned diffusion dynamics admit a finite-dimensional Koopman representation under mild assumptions.

**Questions:**

1. Forgive me if I have missed it, but I would like to see the relationship between the number of pre-constructed noise-data pairs and the distillation performance. One big problem with offline distillation is the possibility of prior holes.

**Ethical Concerns:**

["NO or VERY MINOR ethics concerns only"]

**Final Justification:**

My original main complaint was on the restrictive nature of the setting, and the above response has provided improvement in that aspect. In all, I appreciate the development of the method itself, and offline distillation might be a reasonable setting.

**Limitations:**

Yes.

**Paper Formatting Concerns:**

No.

**Quality:**

2

**Strengths And Weaknesses:**

**Strength**
1. Novel application of Koopman operator theory to the problem of offline diffusion model distillation.
2. The paper provides a solid theoretical foundation for its method. Key contributions include a proof that the learned diffusion dynamics can be represented by a finite-dimensional Koopman operator under mild assumptions, which justifies the core architectural choice.
3. The writing is clear and easy to follow.

**Weakness**
1. The proposed method cannot really utilize more inference-time compute for better generation, unlike Consistency Models.
2. I do not think that the community can agree over the statement on offline distillation methods are more scalable and modular. Generally, the community has moved away from this family of approaches.
3. The empirical results are very weak. The paper only conducts experiments on toyish datasets like CIFAR and FFHQ. The results are lackluster. Without adversarial training, it is worse than the only offline distillation baseline (7.83 vs 6.91). It is not at all comparable to the likes of DMD and CM.

---

> ### Author Rebuttal · Authors · 2025-07-30
>
> Thank you for your constructive and thorough review; it's helped us significantly improve our work. We really appreciate your recognition of the novelty of our Koopman operator-based distillation approach, the strength of our theoretical framework, and the clarity of the paper. We've addressed your questions and concerns below and will incorporate your suggestions into the final revision.
>
> ---
>
> > W1: The proposed method cannot really utilize more inference-time compute for better generation, unlike Consistency Models.
>
> You're right that, in its current form, KDM doesn't support inference-time compute scaling in the same way as Consistency Models. However, we briefly discuss a potential simple extension of this idea in Appendix F. Specifically, since our model is based on the Koopman operator, its infinitesimal version theoretically allows for simulating the full diffusion trajectory via integration. This offers a path to better adhere to the teacher’s dynamics and leverage additional NFEs at inference time. We view this as a promising direction for future work and appreciate you highlighting its relevance.
>
> ---
>
> > W2: I do not think that the community can agree over the statement on offline distillation methods are more scalable and modular. Generally, the community has moved away from this family of approaches.
>
> To avoid vague terminology like "scalable and modular," we'll revise that sentence in the related work section to more clearly communicate the concrete benefits of our approach.
>
> That said, we believe the offline setting remains highly valuable, particularly in scenarios where training without access to the teacher model is desirable, computational and memory resources are limited, or precomputed supervision needs to be reused across multiple student models. These advantages, outlined in Lines 36–42, are especially relevant in resource-constrained environments or where privacy is a key concern. Please also see our response to Reviewer `mkU1` for W1, where we analyze the complexity trade-offs between our method and state-of-the-art online distillation approaches. While the community has indeed focused more on online distillation, we believe that offline distillation offers inherent advantages that can be highly meaningful for many use cases and people in the broader research community.
>
> For instance, from a privacy perspective, offline methods offer a clear advantage by reducing the risk of inversion and membership inference attacks. Unlike traditional online distillation, where the student repeatedly queries the teacher (e.g., during denoising steps), increasing exposure to sensitive information, offline distillation relies on a fixed set of noise–data pairs. Once generated (and optionally anonymized or filtered), the teacher model can be discarded or kept private, significantly reducing the attack surface. This makes it computationally infeasible for either the student or an adversary to recover the original training distribution beyond what is provided. A practical use case of this setting is outsourcing training to third parties without disclosing sensitive model parameters or proprietary data. The data curator (e.g., a hospital or financial institution) can sanitize training samples to remove personally identifiable information (PII), and since the external trainer does not need access to the full generative model or its sampling process, they remain isolated from internal correlations that may otherwise leak private content. This enables privacy-preserving collaboration while retaining model utility.
>
> In general, while our KDM doesn't yet match the performance of leading online techniques, it significantly narrows the gap between offline and online approaches. This reveals the potential of offline methods and highlights a promising, yet underexplored, direction. We hope our work will motivate further research attention in this area, where our results already demonstrate tangible improvements.
>
> ---
>
> > W3: The empirical results are very weak. The paper only conducts experiments on toyish datasets like CIFAR and FFHQ. The results are lackluster. Without adversarial training, it is worse than the only offline distillation baseline (7.83 vs 6.91). It is not at all comparable to the likes of DMD and CM.
>
> Our experimental setup closely follows the standard evaluation protocol established by GET, the most recent and relevant offline distillation baseline. Unfortunately, scaling to large-scale datasets such as ImageNet is currently infeasible for us due to computational resource limitations at our institute. That said, we have incorporated two medium-sized, high-quality datasets, FFHQ and AFHQv2, that were not included in the original GET benchmark, thereby contributing to the expansion of the offline distillation evaluation landscape and showing the models' capability to scale to higher resolution datasets. Within this benchmark, KDM achieves state-of-the-art performance in the offline setting. While our method doesn't yet match the performance of online or simulation-free approaches, we are transparent about this limitation and clearly frame our contributions within the offline distillation paradigm, which has fundamental differences from the online setting.
>
> Additionally, integrating adversarial losses into alternative distillation methods for diffusion models is often non-trivial and presents significant challenges. Many existing approaches lack the architectural flexibility or training stability required to effectively incorporate such objectives, unlike our proposed KDM. Notably, we extended GET with adversarial training, which improved its FID from 6.91 to 6.13. However, KDM still achieves a superior FID of 4.68, demonstrating its effectiveness beyond prior methods. Furthermore, the DMD method is also a method that utilizes a generator and fake vs. real training setup as part of its components, sharing similar outlines to our method. Therefore, we believe it would be somewhat misleading or unfair to exclude the adversarial component, as it's an integral part of our method.
>
> We would also like to highlight that our model is reasonably comparable to Consistency Models (CM). CM achieves an FID of 3.55, while our method attains a score of 4.68, a relatively small margin of just 1.13 FID points. However, CM requires 8 A100 GPUs over several days for training, whereas our method achieves its performance using a single A100 GPU over the same duration, demonstrating significantly greater efficiency. Additionally, the DMD method has only been evaluated on conditional generation with CIFAR-10, achieving an FID of 2.62. On the same task, our model achieves an FID of 3.24, reflecting a small performance gap of only 0.62 points. Therefore, we respectfully disagree with the assertion that our results are not at all comparable. And, we believe that in addition to offering the inherent advantages of offline approaches (e.g., privacy and memory efficiency), it can provide a trade-off between performance and computational cost.
>
> ---
>
> > Q1: Forgive me if I have missed it, but I would like to see the relationship between the number of pre-constructed noise-data pairs and the distillation performance. One big problem with offline distillation is the possibility of prior holes.
>
> We do analyze the effect of the number of precomputed noise-data pairs in our study (see Lines 341–346 and Fig. 5B). As shown there, increasing the dataset size from 250k to 1M samples leads to a substantial improvement in FID, approximately a 33% reduction, highlighting the importance of broad coverage in the training distribution. While we agree that further scaling would provide deeper insight into the limits of offline distillation and potential “prior holes,” we were constrained by available computational resources and leave larger-scale studies to future work.

---

> > ### Comment · Reviewer_FqCX · 2025-08-05
> >
> > Thank you for your detailed response. I have also read other comments and rebuttals. Overall, I remain unconvinced about the validity of imposing such strong constraints against what can and cannot be used during training.
> >
> > About offline distillation: the paper concerns with a very particular setting (offline distillation where the teacher is discarded after generating noise-data pairs) due to a very particular reason (safety against attacks to the teacher, which may reveal sensitive information). First of all, this is not a common setting at all, as I do not see many open-sourced noise-sample paired datasets. In fact, there are many open-source model releases without their training data (for various reasons). If attacking these models and retrieving their training data is easy, I do not think releasing model weights will be as common as it is today. Thus, such a statement needs to be argued in depth. Secondly, even if this is a valid setting (not release model but release their generated noise-data pair), why cannot I just train a diffusion model, then distill with my own trained weights? Better yet, I could train in one stage with IMM or MeanFlow, which will even give me the ability to utilize more inference-time compute for better generation. That is, if the setting is that we only have data but no pre-trained models, then these kinds of approaches become fair game and need to be compared to.
> >
> > About model performance: since none of CM, DMD, and GET utilizes an additional adversarial loss, and given the gap between 7.83 vs. 4.68, it is hard to judge whether the proposed method is actually the core contributor. I want to note that DMD can utilize an additional adversarial loss, which will further boost the performance (see DMD2). DMD was also tested on unconditional CIFAR10, which gives 3.77 (see Appendix E, table 6).
> >
> > Additional comments on cost and efficiency: with offline distillation methods, the overall cost needs to include the cost of generating the noise-data pairs, especially if the performance substantially benefits from a great number of them.

---

> ### Author Response · Authors · 2025-08-06
> **Discussion Cont. - 1**
>
> We would like to thank you for enagging in discussion and helping us to drive forward our work. Below we contuniue the discussion.
>
> > Overall, I remain unconvinced about the validity of imposing such strong constraints against what can and cannot be used during training. ...  the paper concerns with a very particular setting (offline distillation where the teacher is discarded after generating noise-data pairs) due to a very particular reason (safety against attacks to the teacher, which may reveal sensitive information).
>
> We acknowledge that the offline setup [1] involves trade-offs compared to other configurations. Nevertheless, we believe that the focus on results somewhat diverges from the core contribution of our paper.
>
> First, our work presents a novel analysis of diffusion models from the perspective of dynamical systems (Section 4). We demonstrate the emergence of latent structure at the end of the diffusion process through experiments on both toy and real-world datasets (Sections 4 and 6.1). Second, we develop rigorous mathematical foundations and provide non-trivial, novel proofs connecting Koopman theory with diffusion models. These contributions are independent of the offline setup and stand on their own merit. We believe that this new theoretical framework, along with its analysis and practical demonstrations, can be valuable to the community, regardless of the specific application context. We would be happy if our work may won't be judge solely on performance aspects.
>
>
> Furthermore, we would like to clarify that our evaluation approach adheres to [1] to enable a fair comparison. Additionally, we would like to highlight another merit of the offline setup. While privacy is indeed an important advantage, it is not the only potential benefit. Offline methods offer other inherent strengths that are worth considering. For example:
>
> 1. **Model-agnostic distillation**: Offline methods can distill a variety of diffusion models, as demonstrated by our successful distillation of both Flow Matching and EDM using the same framework. In contrast, some online distillation methods are theoretically and practically constrained to specific teacher models.
>
> 2. **Architectural generalization**: Offline approaches can generalize across architectures, from U-Nets to Transformers, whereas some online methods may be more tightly coupled to specific architectures.
>
> 3. **Scalability and flexibility**: Offline methods can scale up or down based on the number of parameters, offering flexibility to meet the needs of downstream applications with different resource or performance constraints. While some other online methods can do it, some other methods rely on inheriting the teacher's weights for the student.
>
> ---
>
>
> > First of all, this is not a common setting at all, as I do not see many open-sourced noise-sample paired dataset
>
> We would like to note that the offline setup was only proposed at the end of 2023 (December) [1], whereas online methods have a considerably longer history, dating back to early 2022 (February) [2]. This nearly two-year gap may introduce a form of historical bias. Additionally, we believe that evaluating scientific contributions based solely on popularity can be misleading. While we acknowledge that online methods currently dominate the field, we think that the offline approach should not be overlooked, as it presents distinct advantages that are complementary to those of online methods.

---

> ### Author Response · Authors · 2025-08-06
> **Discussion Cont. - 2**
>
> >  If attacking these models and retrieving their training data is easy, I do not think releasing model weights will be as common as it is today. Thus, such a statement needs to be argued in depth.
>
> The risk of data theft from open-source models has become a significant and active area of research, as highlighted in works such as [3, 4], among many others. This issue is not unique to image models; it also poses serious challenges in other domains [5] in different contexts.
>
> The current state of openness does not necessarily guarantee safety or long-term sustainability, and it may lead to future challenges, as discussed in these papers and as we also believe. To address your concern, we will incorporate this discussion into the paper to address these argument more thoroughly.
>
> ---
>
> >  Secondly, even if this is a valid setting (not release model but release their generated noise-data pair), why cannot I just train a diffusion model, then distill with my own trained weights? Better yet, I could train in one stage with IMM or MeanFlow, which will even give me the ability to utilize more inference-time compute for better generation.
>
> Regarding the first suggestion, it is a good one; however, it involves multiple steps and is less efficient than leveraging an offline distillation approach. As for your second suggestion, it is also a good idea, however training on generated data may introduce degradation for these models. To ensure a fair comparison, we would need to train models from scratch. To address your comment, **we are making our best effort to incorporate this comparison before the rebuttal deadline** and hope to share the results here in time. Regardless, we will ensure that this comparison is included in the final revision of the paper.
>
> Furthermore, please note that both methods you mentioned (MeanFlow and IMM) are concurrent works, in accordance with the NeurIPS timeline guidelines, and a comparison is therefore not required. Nevertheless, we will do our best to include at least some of it.
>
> ---
>
> > About model performance: since none of CM, DMD, and GET utilizes an additional adversarial loss, and given the gap between 7.83 vs. 4.68, it is hard to judge whether the proposed method is actually the core contributor.
>
> To address your concern, we incorporated adversarial training into GET and successfully adapted it. On CIFAR-10, the adversarially trained GET achieved an FID of 6.13, an improvement over the original GET's 6.91, though it still underperforms compared to our KDM's 4.68.
>
> ---
>
> >Additional comments on cost and efficiency: with offline distillation methods, the overall cost needs to include the cost of generating the noise-data pairs, especially if the performance substantially benefits from a great number of them.
>
> Thank you for raising this important point. In Section 6.2 of our paper, we compare our method to GET in terms of computational complexity, demonstrating that our approach performs favorably among offline methods.
>
> For a more detailed analysis specifically addressing the inclusion of the cost of generating noise–data pairs, we refer to [1], which provides an in-depth investigation of this topic. That work includes the cost of pair generation in its analysis and concludes that offline distillation can be more efficient than online approaches, even when accounting for this overhead. Additionally, it is worth noting that the pair generation is performed only once, after which the overall complexity is reduced.
>
> To improve clarity and address your concern directly, we will incorporate the relevant analysis from [1] into our appendix and reference it from the main text in the final revision.
>
> ---
>
> To conclude, we hope we were able to address the majority of  the concerns you raised and hope to reporting additional results soon.  While we acknowledge that the setup we working in, offline distillation, may appear restrictive and comes with trade-offs, we hope the merits discussed throughout the discussion can demonstrate that it should not be overlooked and given a chance.
>
> We would like to emphasize again that the core contribution of our work lies in the application of Koopman theory. This paper makes fundamental progress both in theoretical grounding and empirical validation. We believe these contributions **stand on their own, independent of the offline setting**, and may offer value to the community even if the offline paradigm is seen as more limited. We hope this perspective, along with our earlier discussion, can help you reconsider your evaluation. Thank you again very much for the constructive feedback.
>
> ---
> [1] One-Step Diffusion Distillation via Deep Equilibrium Models
>
> [2] Progressive Distillation for Fast Sampling of Diffusion Models
>
> [3] Extracting Training Data from Diffusion Models
>
> [4] Membership Inference of Diffusion Models
>
> [5] Scalable Extraction of Training Data from (Production) Language Models

---

> ### Author Response · Authors · 2025-08-07
> **Discussion Cont - 3**
>
> We sincerely thank you for continuing to engage in constructive discussion. Following the previous exchange and in order to address your remaining concerns, we conducted an additional experiment based on your suggestion:
>
> > "... Better yet, I could train in one stage with IMM or MeanFlow."
>
> We trained an IMM model from scratch using only the data (excluding the noise) from the data-noise pairs, and additionally computed the Inception Score (IS), which was not originally reported. Methods were trained under the same compute budget of 102.4 million images. The results are presented below:
>
> | Method                    | FID ↓ | IS ↑ |
> | ------------------------- | ----- | ---- |
> | IMM (with generated data) | 4.81  | 9.17 |
> | GET                       | 6.91  | 9.16 |
> | KDM                       | 4.97  | 9.22 |
> | KDM-F                     | 4.68  | 9.08 |
>
> These results demonstrate that, under the same generated data and budget settings, our method (KDM) is competitive with state-of-the-art approaches for one-step generation from scratch. This supports our claim that KDM is a promising and viable alternative for this task. We also note that, in line with reviewer zaaq's feedback, we included a comparison with a strong baseline in a fair setting (rf-2++), further highlighting the strength of our framework in offline scenarios. As the rebuttal period comes to an end, we are unable to conduct further comparisons at this time. However, given the opportunity, we would be happy to also include a comparison with MeanFlow in the final revision, although its a concurrent work.
>
> Following our in-depth discussion, we will revise the statement in the paper from **state-of-the-art** to **competitive**, as this more accurately reflects the current context after the rebuttal. We sincerely thank you once again for your constructive feedback, which has greatly helped us improve our work.
>
>
> Overall, we hope that the recent discussions, along with the new results, help clarify our contributions, contextualize our method within the broader landscape, and further demonstrate its effectiveness. Once again, we thank you for your thoughtful feedback and remain happy to address any further questions or topics you may have.

---

> > ### Comment · Reviewer_FqCX · 2025-08-09
> >
> > I thank the authors for their detailed follow-up response. I believe that these new additions should be highlighted in the revised paper. Since the current focus is offline distillation (it says so in the title), I do believe that all reasonable baseline approaches should be compared to. Moreover, the motivation part should discuss in-depth the reasons and challenges of such a setting before proposing your method. My original main complaint was on the restrictive nature of the setting, and the above response has provided improvement in that aspect. In all, I appreciate the development of the method itself, and will raise my score.

---

### Note · Authors · 2025-08-12

Dear AC,

We appreciate the opportunity to provide these final remarks and thank the reviewers for their constructive feedback. We summarize below the main concerns and our responses.

**Offline Setting**

FqCX and zaaq raised questions about the limitations of the offline setting. The discussions helped us emphasize that offline methods offer inherent advantages beyond privacy, including model-agnostic distillation, architectural generalization, efficiency and scale flexibility, which online methods do not always provide. We also acknowledge the trade-offs of the offline setup and have incorporated insights from these exchanges to clarify why it should not be overlooked in the distillation landscape. We believe these clarifications largely address the concern, as reflected in FqCX positive feedback.

**GET Baseline**

mkU1 and FqCX noted that GET lacks an adversarial setup. We implemented adversarial GET, improving its FID from 6.91 to 6.13. Our method still achieved a better FID of 4.97.

**Performance**

FqCX raised concerns about the lack of comparisons to one-step generation methods that can work in an offline setup. Following this suggestion, we included a new SOTA baseline, IMM, trained under a setup similar to KDM. Our results show that KDM (FID 4.97, IS 9.22) is highly competitive with IMM (FID 4.81, IS 9.17).

zaaq requested a comparison to rf-2++. First, we clarified the methodological differences after carefully inspecting rf-2++. Then, we ran experiments in a comparable setup, where KDM outperformed rf-2++ and remained competitive in a broader setup.

Based on these findings, we revised our claim from “SOTA” to “competitive,” which more accurately reflects KDM’s standing after the rebuttal phase.

**Training**

dXrx and mkU1 raised concerns about training stability and loss weighting. In response, we added smoothed convergence curves to show stable dynamics, conducted ablations on Koopman and adversarial loss weights confirming robustness, and extracted Koopman norms supporting the relevance of Theorem 5.2.

The above summarizes the key points. Some other issues that were resolved have been omitted due to space constraints. As far as we know, we have addressed all reviewers main concerns. Some gaps like compute-quality trade-off are still open and dedicated for future work. While the discussion with reviewer zaaq remains open, we hope we have addressed their main concerns, as they have not raised further comments following our responses.

---

### Decision · Program_Chairs · 2025-09-17

**Decision:**

Accept (poster)

**Comment:**

This paper leverages Koopman theory to propose a method for diffusion model offline single-function-evaluation distillation. The reviewers unanimously praised the theory, and the connection between Koopman theory and sampling in diffusion models. The main issue reviewers raised was the underperformance of the proposed method as compared to online distillation methods.

Although I agree with the reviewers that outperforming online distillation methods would strengthen the paper, I do not believe such results are necessary for acceptance, as the method performs competitively compared to existing offline distillation approaches, and the presented connections to Koopman theory are valuable to the community.